# Evoked and transmitted culture models: Using bayesian methods to infer the evolution of cultural traits in history

**Alexandre Hyafil**[1]*, **Nicolas Baumard**[2]

**1** Centre de Recerca Matemàtica, Bellaterra, Barcelona, Spain, **2** Institut d'Etudes Cognitives, Ecole Normale Supérieure, Paris, France

* alexandre.hyafil@gmail.com

## Abstract

A central question in behavioral and social sciences is understanding to what extent cultural traits are inherited from previous generations, transmitted from adjacent populations or produced in response to changes in socioeconomic and ecological conditions. As quantitative diachronic databases recording the evolution of cultural artifacts over many generations are becoming more common, there is a need for appropriate data-driven methods to approach this question. Here we present a new Bayesian method to infer the dynamics of cultural traits in a diachronic dataset. Our method called Evoked-Transmitted Cultural model (ETC) relies on fitting a latent-state model where a cultural trait is a latent variable which guides the production of the cultural artifacts observed in the database. The dynamics of this cultural trait may depend on the value of the cultural traits present in previous generations and in adjacent populations (*transmitted culture*) and/or on ecological factors (*evoked culture*). We show how ETC models can be fitted to quantitative diachronic or synchronic datasets, using the Expectation-Maximization algorithm, enabling estimating the relative contribution of vertical transmission, horizontal transmission and evoked component in shaping cultural traits. The method also allows to reconstruct the dynamics of cultural traits in different regions. We tested the performance of the method on synthetic data for two variants of the method (for binary or continuous traits). We found that both variants allow reliable estimates of parameters guiding cultural evolution, and that they outperform purely phylogenetic tools that ignore horizontal transmission and ecological factors. Overall, our method opens new possibilities to reconstruct how culture is shaped from quantitative data, with possible application in cultural history, cultural anthropology, archaeology, historical linguistics and behavioral ecology.

## Introduction

### Evoked and transmitted culture

It is standard in behavioral sciences to distinguish between evoked culture and transmitted culture [1–5]. Transmitted culture refers to behaviors and beliefs that are mostly due to cultural transmission, either vertically from previous generations (parents, teachers, elders) or

**Data Availability Statement:** Codes are available on the following public repository: https://github.com/ahyafil/Evoked_Transmitted_Culture.

**Funding:** This research was supported by the Spanish State Research Agency (RYC-2017-23231

and CEX2020-001084-M to A.H.) and the Agence Nationale pour la Recherche (ANR-17-EURE-0017 Frontcog, ANR-10-IDEX-0001-02 PSL* to N.B.).

**Competing interests:** The authors have declared that no competing interests exist.

horizontally from peers and neighbors. Language is a typical example of transmitted culture as the best predictor of an individual's language is the language of her parents. By contrast, evoked culture refers to behaviors and beliefs that are mostly due to the expression of an evolved behavioral program in response to some environmental cues. A typical example here is the number of children per woman [6–8]: the environment in which this individual lives (in terms of income, social support, interpersonal violence) is an important indicator of an individual's reproductive preference (although transmitted norms also probably play a role [9]). For example, in response to economic development, most middle-income countries (such as Iran or Korea) have experienced a rapid transition from 6 children per woman to 2 children per woman, regardless of their original religious tradition and socio-moral norms [10, 11].

Obviously, transmitted and evoked cultures are located on the same continuum. All evoked cultural behaviors are also partially influenced by cultural transmission. For instance, an evolved program may trigger an early onset of sexual debut, but the courtship involved in finding a mate may be heavily influenced by transmitted traditions (e.g. the 'date' tradition in the 20th c. US culture). In the same way, all transmitted behaviors are also partially influenced by evoked factors. For instance, phonology has recently been shown to be influenced by dietary and behavioral practices through changes in bite configuration [12]. Even the content of artistic works can be shaped by the environment. For instance, it has been shown that people living in more favorable environments tend to be more trustworthy and to invest more in cooperation. In line with this observation, individuals in European portraits tend to display more pro-social signals in more economically developed periods in history traditions [13]. Here we are interested in quantifying the relative influence of evoked and transmitted forces onto cultural evolution.

## Computational tools for assessing models of cultural evolution

Disantangling evoked and transmitted factors is often difficult. Variation between any two cultures may result from separate traditions which lead to the transmission of two different transmitted cultures or from different local environments which trigger different behavioral programs leading to different evoked cultures. Conversely, uniformity between two cultures could result from similar traditions coming from the same source or from the triggering of similar evolved mechanisms in similar environments.

Typical quantitative cultural evolution studies use a phylogenetic method to reconstruct when a particular trait emerged in the past [12, 14–16]. Phylogenetic methods are however limited by the differences between cultural and genetic evolution. These methods do not take into account horizontal tranmission of cultural traits, i.e. the geographical diffusion of culture by communication between populations. Moreover, genetic evolution does not include any evoked component, i.e. the role of ecological factors such as socio-economic development, latitude, or the size of the population. A possibility to take all these factors into account is to use instead a regression method [12, 17] that incorporates the influence of ecology, shared cultural background and spatial proximity onto the observed cultural trait. However regression methods may not be as accurate as methods that rely on an explicit model of cultural evolution through time, and are necessarily arbitrary in what they define as shared cultural background. Importantly, they do not permit to reconstruct the evolution of a cultural trait through time and across regions.

## Cultural evolution with diachronic datasets

Another important characteristic of the studies mentioned above is that cultural traits as well as ecological factors are observed only once (*synchronic dataset*), usually at the time of the

study. However, the development of long-term series of socio-economic indicators [18–21], combined with quantitative records of cultural traits and cultural productions (*diachronic dataset*) [22, 23], now makes it possible to directly assess the impact of ecological and transmitted factors in cultural evolution. To address these limitations, we describe a general statistical framework to infer the specific causes of cultural evolution in a diachronic or synchronic dataset. The framework relies on a generative model for the evolution of cultural traits that incorporates evoked components and spatial diffusion.

## Materials and methods

### Defining ETC models

We first distinguish between cultural *traits* and cultural *artifacts*. Cultural artifacts correspond to the *material production* of a given population: tools, songs, portraits, watches, novels, etc. (see Fig 1). By contrast, cultural traits are the *cognitive and behavioral dispositions* shared by the member of the population: social trust, belief in a moralizing god, conservativeness, etc. Recent developments in cognitive sciences suggest that it is possible to infer cultural traits

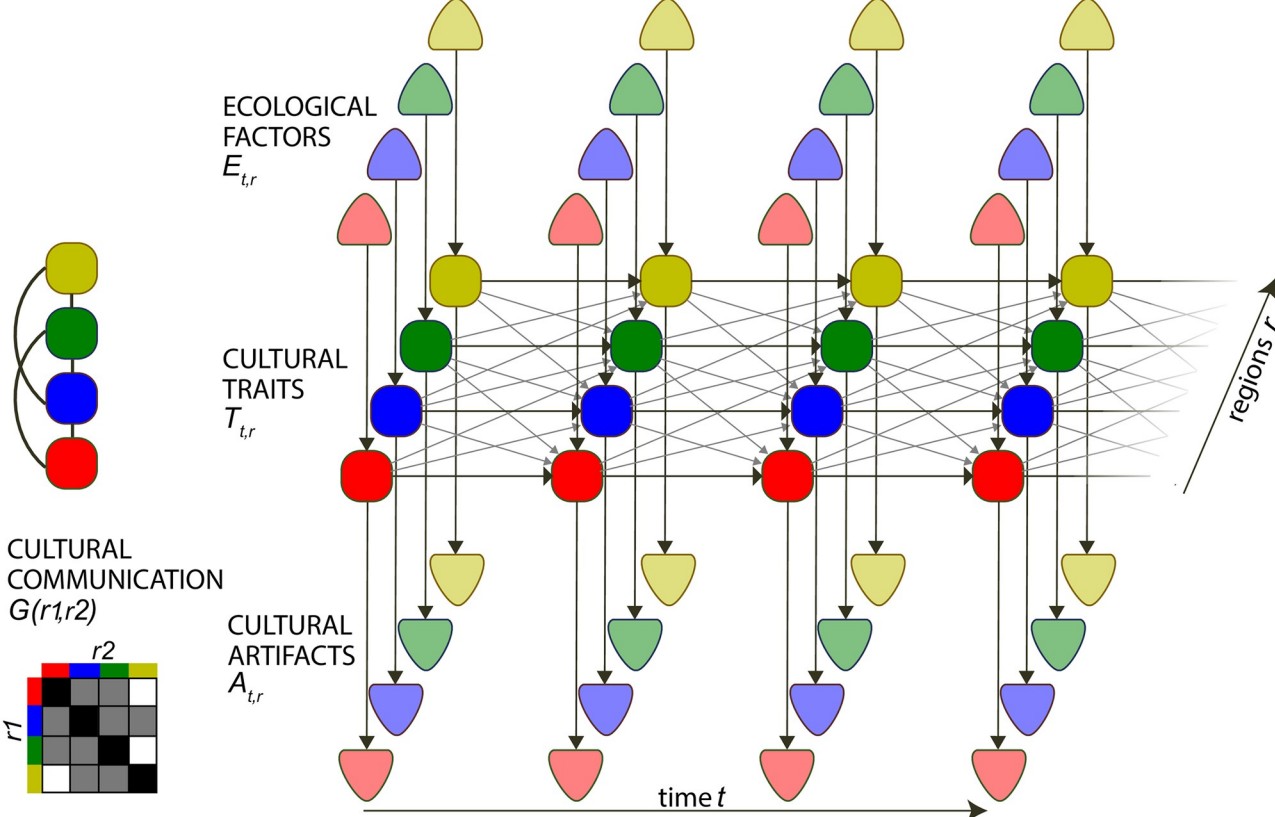

**Fig 1. Evoked and Transmitted Culture (ETC) model.** ETC can be represented as a probabilistic directed graph [38], where nodes $E_{tr}$ and $A_{tr}$ are observed, while nodes $T_{tr}$ are unobserved. Cultural traits $T_{tr}$ evolve through time in a set of interconnected regions (here 4, depicted by different colours). Horizontal arrows represent how cultural trait is inherited from previous time step in same region, while diagonal arrows represent diffusion from interconnected regions (cultural communication). Vertical arrows pointing to cultural traits node represent the influence of ecological factors on cultural trait, while vertical arrows pointing from cultural trait represent the observation process, i.e. how cultural artifacts observed at a specific time $t$ in region $r$ depends on the cultural trait at the same time and location. Left inset: connectivity between different regions, where each edge represent the existence of cultural communication between two regions ($G$ can also take continuous values, i.e. scale inversely to the distance between regions). The information is summarized in the connectivity matrix $G$ below.

from cultural artifacts. For instance, facial action unites in portraits provide information about the social traits (trustworthiness, dominance) sitters want to display [24]; the minor mode, a set of tones that is generally associated with subdued, sad or dark emotions [25] and enjoyed preferably by individuals high on empathy [26, 27], provides information about the personality traits of the audience; imaginary worlds in fictions is associated with higher openness-to-exploration and preferences for explorations [28, 29].

In line with these developments, recent works have used a diversity of cultural artifacts such as portraits [13], theater plays [30], music [31] and movies [32] to infer the evolution of cultural traits such as social trust [13, 30], positive and negative moods [33], wellbeing [31, 34], individualism [35, 36], romantic love [37] and exploratory preferences [32]. In all these cases, scientists build on cognitive and behavioral sciences to connect a specific cultural aspect of artifact (the smile in portrait) with the underlying cultural trait (the priority given to appearing trustworthy), and then reconstruct the long-term evolution of this cultural trait using a long-term series of cultural artifacts.

Cultural traits can be influenced by several factors: they can be transmitted from generation to generation (vertical cultural transmission), transmitted from one society to its neighbor (horizontal cultural transmission) or evoked by ecological factors (socio-economic development, political organization, etc.).

We will use the following example as an illustration: in this example, the cultural trait is the general support for gender equity in the population (a cognitive disposition), while the cultural artifacts could be a corpus of fictions, and for each fiction in the corpus we measure (quantitatively) the importance of female characters. We expect that the cognitive disposition at a given time and region will influence the content of the artifacts: the more support there is for gender equity in the population, the more likely we expect to find important female characters in the fictions produced at that time and region. A general question in cultural evolution is to understand the relative importance of ecological factors, vertical transmission and horizontal transmission in the evolution of the cultural trait. Because we have no direct access to the mental dispositions of ancient populations, we want to infer the dynamics of the cultural traits using the indirect observations of cultural artifacts. In our example, we would want to understand what drives changes in the support for gender equity in the population from a dataset of fiction books at different times and regions, each classified according to the importance of its female characters. If our hypothesis is that economic development leads to higher support of gender equity, we could use the GDP per habitant (a proxy to economic development) as the ecological factor.

We formalize this problem under the generic name of Evoked and Transmitted Culture (ETC) models, using the general formalism of probabilistic generative models. A probabilistic generative model refers to a parametric model which defines a probability distribution for the observations (here, the cultural artifacts) when the input variables of the models (here, the number of regions and ecological variables) are known. Bayesian methods allow notably to estimate the parameters of the generative model that best fit the observations in the dataset (e.g. through Maximum Likelihood Estimation) and compare which of a set of models is best supported by the observations. In our example, we could compare a model where economic development is taken as the ecological factor driving changes in support for gender equity with a model where the average level of science education is the ecological factor. ETC models are special forms of latent process models (also called state-space or Hidden-Markov models), where an unobserved (latent) process evolves through time and determines the value of some observed variables [38]. Here in general, we take a set or $K$ interconnected geographical regions and want to understand the cultural evolution in each of these regions within a certain period of time. For practical reasons we use a set of $n$ discrete time points with time step $dt$.

The cultural trait takes a certain value $T_{t,r}$ for each region $r$ and each time point $t$, and its evolution is guided by its value at the preceding time step, as well as the value in neighbouring regions $\mathcal{N}(r)$ and the value of ecological factors $E_{t,r}$:

$$p(T_{t,r}|\boldsymbol{T}_{t-1}, E_{t,r}) = f((T_{t,r}, T_{t-1,r}, \{(T_{t-1,s}, G_{r,s}), s \in \mathcal{N}(r)\}, E_{t,r}; \boldsymbol{\theta}_T)) \tag{1}$$

$G_{r,s}$ represents the connectivity between regions $r$ and $s$ in the set of regions $\mathcal{N}(r)$, i.e. $G_{r,s}$ is the level of cultural communication between the two regions. $\boldsymbol{\theta}_T$ represents a set of parameters of cultural evolution (which control notably the level of horizontal transmission, vertical transmission and evoked component, see details in following sections). The overall communication between regions is summarized in the cultural connectivity matrix $\boldsymbol{G}$, which here is taken to be constant in time. It is straightforward to extend the framework to use instead a time-varying connectivity $\boldsymbol{G}(t)$, for example to account for the opening and closing of channels of communication between regions. One important implication of Eq 1 is that the evolution of cultural traits follow a Markov process, i.e. cultural traits depend on the past only through their value at the previous time step.

Defining the temporal limits and the borders of a specific 'culture' is tricky. What count as a culture? What counts as a new culture? For pragmatic reasons, we will consider two cases for creation of a new culture. Either it is created *de novo* (from populations for which we do not have previous records in the database), or it is the offspring of another region (i.e. when a population moves en masse to the new area or split into different populations whose cultural dynamics will now diverge). In case of a region created *de novo*, we will assume that the value of the cultural trait for the new region at its time of creation $t_0$ is sampled from some initial distribution $p(T_{r,t_0}) = p_0(T_{r,t_0})$. In the case of an offspring region, we will assume that the value of the new region $r$ is directly transferred from its parent region $s$, i.e. $T_{r,t} = T_{s,t}$.

The production of artifacts is guided by the value of the cultural traits, and possibly as well by some other (known) factors $F_{r,t}$ that can relate to the material conditions of artifact productions in region $r$ at time $t$ (e.g. the number of fiction books published) and to the data collection process (e.g. the records of ancient publications):

$$p(A_{t,r}) = g(T_{t,r}, F_{t,r}; \boldsymbol{\theta}_A) \tag{2}$$

where $\theta_A$ are the parameters of the artifact production process. For example, if each cultural artifact is a binary variable, then we could define $p(A_{t,r}) = S(\beta_0 + \beta_1 T_{t,r} + \beta_2 F_{t,r})$ where $S$ is the logistic function and $\theta_A = \{\beta_0, \beta_1, \beta_2\}$ are parameters that control the bias and the influence of the trait and factor on the cultural artifact, respectively.

While in general we want to infer the trajectory of unobserved cultural traits from the observations of cultural artifacts, it is also possible that the values of cultural traits are accessible at some moment (typically at contemporary times, i.e. at the final time step), and we can include such information to further constrain the ETC model.

In the following sections we discuss two distinct classes of ETC models: when the cultural trait takes the form of a binary value ($T_{t,r} \in \{0, 1\}$), and when it takes the form of a continuous value ($T_{t,r} \in \mathbb{R}$). In our example, these two classes correspond to classifying a population as supporting or non-supporting gender equity for the binary trait; or to quantifying the support to gender equity in the population with a continuous value.

## Binary cultural traits

In many circumstances it is convenient to model a cultural trait as a binary value ($T_{t,r} \in \{0, 1\}$), such as the presence or absence of the trait in the population [22]. In this case the probability of trait transitions between the two consecutive time steps $t$ and $t + dt$ is defined by time-

dependent rates of transitions $R_{\text{up}}(t, r)$ and $R_{\text{down}}(t, r)$:

$$
\begin{aligned}
p(T_{t+1,r} = 1 | T_{t,r} = 0) &= R_{\text{up}}(t, r)dt \\
p(T_{t+1,r} = 0 | T_{t,r} = 0) &= 1 - R_{\text{up}}(t, r)dt \\
p(T_{t+1,r} = 0 | T_{t,r} = 1) &= R_{\text{down}}(t, r)dt \\
p(T_{t+1,r} = 1 | T_{t,r} = 1) &= 1 - R_{\text{down}}(t, r)dt
\end{aligned}
\tag{3}
$$

Here the approximation is valid only if both rates are much smaller at each time than the inverse of the time step ($R_{\text{up}}(t, r), R_{\text{down}}(t, r) \ll 1/dt \, \forall (t, r)$). If the time step is too large, then in Eq 3 the probability of transition in one time step is not infinitesimal, so the transition rates $R_{\text{up}}(t, r)$ and $R_{\text{down}}(t, r)$ cannot be approximated by a constant during each time step due to their dependence in cultural traits (see below).

Initial values are parametrized by $p(T_{1,r} = 1) = x_0$ (and also for regions created *de novo*), with $x_0$ the probability that the cultural trait is present in any new region. The rates of transitions depend on the environmentally shaped susceptibility of innovation (transition from trait absent to trait present) and loss of trait (transition from trait present to trait absent), as well as on the diffusion from neighbouring areas. For the sake of simplicity and interpretability of the model, we propose here linear effects, i.e. the transitions rates depend linearly on the value of the ecological factor and cultural traits in neighbouring regions:

$$
\begin{aligned}
R_{\text{up}}(t, r) &= b_{\text{up}} + \lambda E_{t,r} + \xi \Big( \sum_{s \in \mathcal{N}(r) | T_{t,s}=1} G_{r,s} - \sum_{s \in \mathcal{N}(r) | T_{t,s}=0} G_{r,s} \Big) \\
R_{\text{down}}(t, r) &= b_{\text{down}} - \lambda E_{t,r} + \xi \Big( \sum_{s \in \mathcal{N}(r) | T_{t,s}=0} G_{r,s} - \sum_{s \in \mathcal{N}(r) | T_{t,s}=1} G_{r,s} \Big)
\end{aligned}
\tag{4}
$$

where $b_{\text{up}}$ is the intrinsic rate of cultural innovation, and $b_{\text{down}}$ is the intrinsic rate for loss of the cultural trait, $\lambda$ is the sensitivity of the cultural transition to the ecological factor and $\xi$ determines the rate of cultural diffusion. These five parameters $\boldsymbol{\theta}_T = (x_0, b_{\text{up}}, b_{\text{down}}, \lambda, \xi)$ determine the dynamics of the cultural traits. Eq 4 can be rewritten in terms of the multivariate rate of transitions for all regions $\boldsymbol{R}_{\text{up}}(t)$ and $\boldsymbol{R}_{\text{down}}(t)$:

$$
\begin{aligned}
\boldsymbol{R}_{\text{up}}(t) &= b_{\text{up}} \mathbf{1} + \lambda \boldsymbol{E}_t + \xi \boldsymbol{G}(2\boldsymbol{T}_{t-1} - \mathbf{1})' \\
\boldsymbol{R}_{\text{down}}(t) &= b_{\text{down}} \mathbf{1} - \lambda \boldsymbol{E}_t - \xi \boldsymbol{G}(2\boldsymbol{T}_{t-1} - \mathbf{1})'
\end{aligned}
\tag{5}
$$

Combining Eqs 3 and 5 allows to compute the time-dependent transition rate $p(\boldsymbol{T}_{t+1} | \boldsymbol{T}_t; \boldsymbol{\theta}_T)$. This system is equivalent to an input-output Hidden Markov Model (HMM) [39] where the latent variable is defined as the combination of the cultural trait for all regions, thus taking $2^K$ possible values: $\boldsymbol{T}_t \in \{0, 1\}^K$. For large number of regions $K$ (more than 10–20), the number of combinations becomes prohibitive, so we must resort to approximate solutions using variational inference or sampling methods [40].

## Continuous cultural traits

Here we present the case when the trait takes a continuous value instead of binary value. We propose a simple stochastic linear model for the dynamics of cultural traits, where the degree of vertical transmission, horizontal transmission and evoked component (i.e. influence of the ecological variable) are controlled by separate parameters (to be estimated from the data). We define the dynamics of the cultural trait as following the linear influence of the ecological

factor and value of the trait in neighbouring regions, as captured in the following differential equation (in continuous time)

$$\frac{dT_{t,r}}{dt} = -\rho T_{t,r} + \gamma + \lambda E_{t,r} + \xi \sum_{s \in \mathcal{N}(r)} G_{r,s}(T_{ts} - T_{t,r}) + \sigma \eta \tag{6}$$

Parameter $\rho$ is the *cultural lability* (or cultural leak). It determines the time scale at which the memory of cultural traits is lost, in other words it controls the degree of vertical transmission. For example, in an isolated region (no neighbour), in the absence of ecological variable and noise, then $T_{t,r}$ converges exponentially to $\gamma/\rho$ at time scale $1/\rho$, i.e. the cultural trait is "forgotten" with a time scale of $1/\rho$. $\gamma$ is a bias towards positive or negative trait values capturing a "default" tendency of the cultural trait in the absence of evoked component and horizontal transmission. Because the raw value of cultural traits is usually arbitrary and one is usually mostly interested in relative values, one can often assume that $\gamma = 0$, i.e. that the "default" trait value is null. If this is not the case, one must take caution in defining the generative process for artifacts to make sure that a bias term is not present that would make parameters not identifiable. For example, for binary artifacts, one could use $p(A_{t,r}) = S(\beta_0 + \beta_1 T_{t,r})$ if $\gamma$ is assumed to be null or $p(A_{t,r}) = S(\beta_1 T_{t,r})$ otherwise; the two systems are mathematically equivalent, they only differ in whether the constant bias $\beta_0$ is absorbed into a fixed offset of cultural traits or not. $\eta$ is a Wiener process capturing the stochastic component in cultural evolution, and $\sigma$ is the noise parameter. As for the binary case, $\lambda$ represents the susceptibility to the ecological variable, and $\xi$ the rate of cultural diffusion. Eq 6 can be viewed as an Ornstein-Uhlenbeck process where the drift term depends linearly on the value of the ecological variable and cultural traits in neighbouring regions, while the diffusion term is constant. Eq 6 is transformed in discrete time as a linear-Gaussian evolution equation, taking the form:

$$T_{t,r} = (1 - \rho dt)T_{t-1,r} + dt(\gamma + \lambda E_{t,r} + \xi \sum_{s \in \mathcal{N}(r)} G_{r,s}(T_{t-1,s} - T_{t-1,r})) + \sigma\sqrt{dt}\eta \tag{7}$$

The cultural trait is initiated independently at each region from a gaussian distribution centered on $x_0$ and of variance $\sigma_0^2$. The same applies to regions created *de novo*, while regions created from a parent simply inherit the value of the trait for the parent region.

Eq 7 can be reframed in terms of the $K$-dimensional vector $\boldsymbol{T}_t$:

$$\boldsymbol{T}_t = (1 - \rho dt)\boldsymbol{T}_{t-1} + dt(\gamma \boldsymbol{1} + \lambda \boldsymbol{E}_t + \xi \tilde{\boldsymbol{G}} \boldsymbol{T}_{t-1})) + \sigma\sqrt{dt}\boldsymbol{\eta}, \boldsymbol{\eta} \sim \mathcal{N}(\boldsymbol{0}; \boldsymbol{I}) \tag{8}$$

where $\tilde{\boldsymbol{G}}$ is defined by $\tilde{G}_{rr} = G_{rr} - \sum_s G_{rs}$ and $\tilde{G}_{rs} = G_{rs}$ for $r \neq s$.

Thus $\boldsymbol{T}$ is a multivariate auto-regressive model of order one, its dynamics is given by:

$$\begin{cases} \boldsymbol{T}_t &= \boldsymbol{M}\boldsymbol{T}_{t-1} + \boldsymbol{J}_t + \boldsymbol{\eta}\sqrt{dt}, \text{ with} \\ \boldsymbol{M} &= (1 - \rho dt)\boldsymbol{I} + \xi dt\tilde{\boldsymbol{G}} \text{ and} \\ \boldsymbol{J}_t &= (\lambda \boldsymbol{E}_t + \gamma \boldsymbol{1})dt \end{cases} \tag{9}$$

The initial state is $\boldsymbol{T}_0 \sim \mathcal{N}(x_0\boldsymbol{1}, \sigma_0^2\boldsymbol{I})$.

The set of parameters for the cultural trait evolution is $\boldsymbol{\theta}_T = (\gamma, \lambda, \xi, \rho, \sigma^2, x_0, \sigma_0^2)$. The first five parameters relate to the cultural trait evolution $\boldsymbol{\theta}_{Te})$, while the last two refer to the initialization ($\boldsymbol{\theta}_{Ti}$). Depending on the particular ETC problem, there may also be parameters associated with the artifact production model, that we denote generically $\theta_A$.

## General fitting procedure

We are generally interested in estimating from a dataset (including cultural artifacts $\mathbf{A}$, ecological factors $\mathbf{E}$ as well as the cultural connectivity $\mathbf{G}$) the set of parameters $\boldsymbol{\theta} = (\boldsymbol{\theta}_T, \boldsymbol{\theta}_A)$ from the ETC model. Here we define $\mathbf{A}$ and $\mathbf{T}$ as the entire collection of cultural artifacts and traits, i.e. $\mathbf{A} = (\mathbf{A}_1..\mathbf{A}_n)$ and $\mathbf{T} = (\mathbf{T}_1..\mathbf{T}_n)$. Best fitting parameters report the degree of ecological determination, vertical and horizontal transmission of the cultural traits, and allow estimating the dynamics of the cultural traits across time. They also allow comparing different models, for example to assess which ecological factor accounts for the variation of cultural traits.

There is in general no analytical form to find a solution to this problem, except for some cases such as HMM (see Appendix A in S1 Appendix) or linear dynamical systems (i.e. continuous trait variable with continuous artifacts generated as gaussian observations). For other types of model, we will have to recur to approximate solutions, using a variety of techniques such as analytical approximations, variational approaches or sampling methods [38]. Most of these methods rely on variations of the Expectation-Maximization (EM) algorithm (although see [41, 42] for alternative approaches). The EM algorithm runs recursively between two stages. In the Expectation stage, we compute the posterior probability for the value of the (unobserved) cultural traits, given the (observed) cultural artifacts and the current value of the parameters, i.e. $q(\mathbf{T}; \boldsymbol{\theta}) = p(\mathbf{T}|\mathbf{A}, \boldsymbol{\theta})$. In case the value of cultural traits are observed at some points, inference will also be based on these observed traits $\mathbf{T}^{\mathrm{obs}}$: $q(\mathbf{T}_t; \boldsymbol{\theta}) = p(\mathbf{T}_t|\mathbf{A}, \mathbf{T}^{\mathrm{obs}}, \boldsymbol{\theta})$. While exact inference can be performed when the cultural trait is binary (see Appendix A in S1 Appendix), it cannot in general when the cultural trait is continuous, so we have to resort to approximate methods to evaluate the posterior $q(\mathbf{T}_t; \boldsymbol{\theta})$. We have investigated three types of approximations: moment method, Laplace approximation and Expectation-Propagation (EP) (see Appendix B in S1 Appendix for details).

The second stage of the EM algorithm is the Maximization stage. In this stage, we look for the parameters $\boldsymbol{\theta}$ that best account for the observed data, given the estimation of the value of cultural artifacts from the E-step, i.e we maximize $Q(\boldsymbol{\theta}; \boldsymbol{\theta}^{old}) = \int q(\mathbf{T}; \boldsymbol{\theta}^{old})\log p(\mathbf{A}, \mathbf{T}|\boldsymbol{\theta})d\mathbf{T}$ w.r.t. $\boldsymbol{\theta}$:

$$
\begin{aligned}
Q(\boldsymbol{\theta}; \boldsymbol{\theta}^{old}) &= \int q(\mathbf{T}; \boldsymbol{\theta}^{old})\log p(\mathbf{A}, \mathbf{T}|\boldsymbol{\theta})d\mathbf{T} \\
&= \int q(\mathbf{T}; \boldsymbol{\theta}^{old})\log p(\mathbf{T}|\boldsymbol{\theta}_T)d\mathbf{T} + \int q(\mathbf{T}; \boldsymbol{\theta}^{old})\log p(\mathbf{A}|\mathbf{T}, \boldsymbol{\theta}_A)d\mathbf{T}
\end{aligned}
\tag{10}
$$

We define $Q_T(\boldsymbol{\theta}_T)$ and $Q_A(\boldsymbol{\theta}_A)$ as resp. the first and second integrals in the right hand side. We can see that trait parameters $\boldsymbol{\theta}_T$ and production parameters $\boldsymbol{\theta}_A$ can be estimated independently by maximizing $Q_T$ and $Q_A$ separately. Since trait evolution is a Markov process, $p(\mathbf{T}) = p(\mathbf{T}_0)\Pi_t \, p(\mathbf{T}_t|\mathbf{T}_{t-1})$, $Q_T$ can further be dissociated into initialization and evolution terms:

$$
\begin{aligned}
Q_T(\boldsymbol{\theta}_T; \boldsymbol{\theta}^{old}) &= \int q(\mathbf{T}_0)\log p(\mathbf{T}_0|\boldsymbol{\theta}_{Ti})d\mathbf{T} + \sum_{t=1}^n \int q(\mathbf{T}_t, \mathbf{T}_{t-1})\log p(\mathbf{T}|\mathbf{T}_{t-1}, \boldsymbol{\theta}_{Te})d\mathbf{T} \\
&= Q_{Ti}(\boldsymbol{\theta}; \boldsymbol{\theta}^{old}) + Q_{Te}(\boldsymbol{\theta}_{Te}; \boldsymbol{\theta}^{old})
\end{aligned}
\tag{11}
$$

$Q_A(\boldsymbol{\theta}_A)$ can also be decomposed into a sum of terms depending on each artifact independently:

$$
\begin{aligned}
Q_A(\boldsymbol{\theta}_A; \boldsymbol{\theta}^{old}) &= \int q(\mathbf{T})\log p(\mathbf{A}|\mathbf{T}, \boldsymbol{\theta}_A)d\mathbf{T} \\
&= \sum_{t,r} \int q(T_{t,r})\log p(A_{t,r}|T_{t,r}, \boldsymbol{\theta}_A)dT_{t,r}
\end{aligned}
\tag{12}
$$

The algorithm stops when the change in log-likelihood log $\mathcal{L}(\boldsymbol{\theta}) = \log\ p(\boldsymbol{A}|\boldsymbol{T}, \boldsymbol{\theta})$ of the parameters between two successive iterations is below a certain threshold. Since the (log-) likelihood function is in general non-convex, the EM procedure can converge to a local (non global) minimum. To mitigate this problem, multiple EM loops are launched, each with a different set of initial values for parameters $\boldsymbol{\theta}$. The set of final parameters with overall higher likelihood is retained.

Once we have identified best-fitting parameters $\hat{\boldsymbol{\theta}}$, we can infer the trajectory of cultural traits across time in the different region, which is given directly by the result of the E-step $q(\boldsymbol{T}; \hat{\boldsymbol{\theta}})$. We also define measures of influence, which compare how much variance in the cultural trait value can be explained by the value of the cultural trait in the same region one generation before (*influence of vertical transmission*), by the value of the cultural trait in neighbouring regions (*influence of horizontal transmission*) and the value of the ecological factor in the same region (*influence of environment*, see Appendix B5 in S1 Appendix for details).

## Confidence interval and model comparison

Once we have obtained the best-fitting values $\hat{\boldsymbol{\theta}}$ for the ETC parameters, we are generally interested in deriving some confidence intervals for these values. We will use two different ways to compute them. Since in a latent model observations are not independently drawn, classical bootstrapping methods cannot be used; we can use parametric bootstrapping instead [43]. In parametric bootstrapping, we evaluate the dispersion in parameter estimation that we should expect if the true model corresponded indeed to the ETC model with best-fitted values. We will generate $n_B$ different samples from the ETC parametrized with best fitting value, i.e. we sample from $\boldsymbol{A}|\hat{\boldsymbol{\theta}}$. Each sample has the exact same number of observations at each data point as in the real dataset. Then the fitting procedure is applied to each of the synthetic datasets. The distribution $\boldsymbol{\theta}_{\text{bootstraps}}$ of the parameters reflects the intrinsic uncertainty of the estimation process for our dataset. It can be used directly to extract confidence interval for each parameter.

An alternative strategy is to use the Laplace approximation to approximate the posterior distribution over parameters $p(\boldsymbol{\theta}|\boldsymbol{A})$ by a Gaussian distribution centered on the MAP parameters $\mathcal{N}(\hat{\boldsymbol{\theta}}, \Sigma)$ [38]. Posterior distribution for each parameter is the approximated by $\mathcal{N}(\theta_i, \Sigma_{ii})$, and significance for the parameter vs. null hypothesis $\theta_i = 0$ can be tested using the Wald test. $\Sigma$ is taken to be the inverse of the Hessian of $-\log\ \mathcal{L}(\boldsymbol{\theta})$ taken at the MAP parameters, i.e. it is defined by the local curvature of the log-likelihood function.

Following [44], the Hessian $\boldsymbol{H}$ of the log-likelihood function evaluated through an EM algorithm can be computed following

$$\boldsymbol{H} = \nabla\nabla \log\ \mathcal{L}(\boldsymbol{\theta}) = \nabla\nabla Q(\hat{\boldsymbol{\theta}})\,(\boldsymbol{I} - \nabla(arg\,max_{\boldsymbol{\theta}}Q(\boldsymbol{\theta}; \boldsymbol{\theta}^{old})|_{\boldsymbol{\theta}^{old}=\hat{\boldsymbol{\theta}}}) \tag{13}$$

The Jacobian can be evaluated by methods of finite differences, while the Hessian of $Q$, because of the decomposition of $Q$ in Eq 11, decomposes also into Hessians related to $Q_T$ and $Q_A$:

$$\nabla\nabla Q(\hat{\boldsymbol{\theta}}) = \begin{bmatrix} \nabla\nabla Q_T(\hat{\boldsymbol{\theta}}_T) & \mathbf{0} \\ \mathbf{0} & \nabla\nabla Q_A(\hat{\boldsymbol{\theta}}_A) \end{bmatrix} \tag{14}$$

Both Hessians can be computed analytically in a variety of cases (see Appendices A2 and B2 in S1 Appendix).

Model comparison permits to compare the full version of the model to reduced versions, where either diffusion between regions and/or evoked component is removed. To perform model comparison, we can use the log model evidence (marginalized over parameter vlaues), which can be approximated using the Laplace approximation (and assuming a broad prior for parameters $\theta$) by:

$$\log p(\boldsymbol{A}) \approx \log p(\boldsymbol{A}|\hat{\boldsymbol{\theta}}) + 1/2\log|H| \tag{15}$$

Other methods that penalize the log-likelihood based on the numbers of parameters such as the Akaike Information Criterion (AIC) or Bayesian Information Criterion (BIC). Alternatively, model comparison can be performed using cross-validation, which is more expensive computationally but will be more accurate when the Laplace approximation is not appropriate, notably when there are multiple local maxima in the LLH. Here we used k-fold cross-validation, where model performance was assessed using Cross-Validated Log-Likelihood.

## Results

### Testing with binary ETC model

We first tested our estimation method on synthetic data generated from an ETC with binary traits and binary artifacts. The model was run over 6 interconnected regions over 1000 time steps (see details in Appendix C1 in S1 Appendix). We compared the true value with the estimated values for cultural traits and parameters of the model (Fig 2a). The fitting algorithm captured the overall dynamics of the up and down states of the cultural traits in all regions. It also provided reliable estimates of the parameters of the ETC model (Fig 2b). Finally we performed Bayesian model comparison of the full ETC model with reduced model where we either removed horizontal transmission (*evoked-only model*, where we enforced $\xi = 0$) or evoked component (*diffusion-only model*, where we enforced $\lambda = 0$). BIC was larger in either reduced model than in the full model ($\Delta BIC \approx 15$), showing that the Bayesian procedure

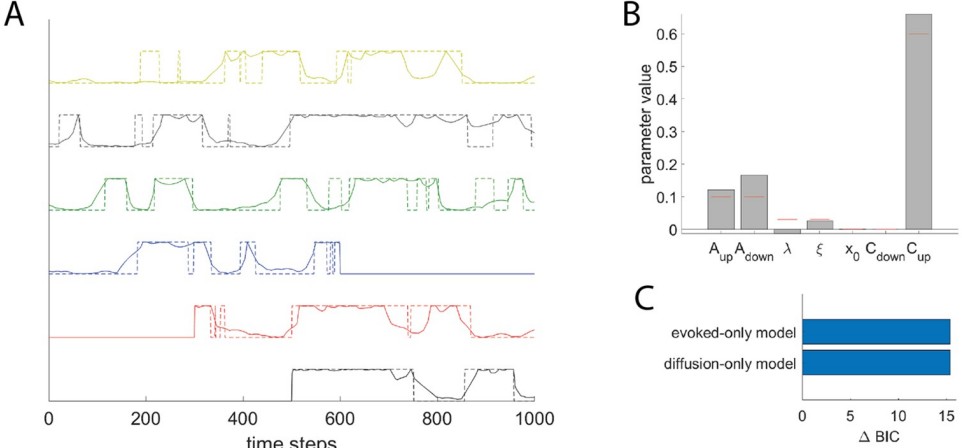

**Fig 2. Fitting a binary ETC model on synthetic data. A**. True vs. inferred cultural traits across time in all regions (bottom to top, from region 1 to region 6). Full lines represent the true binary cultural trait $T_{t,r}$, while dotted line represent the posterior probability of the cultural trait given by estimated parameters $p(T_{t,r}|\hat{\boldsymbol{\theta}})$, where parameters are estimated using the EM procedure from observed data (**E**, **A**). **B**. Value of parameters estimated with the EM procedure (gray bars) against true value of the parameters (red ticks). **C** Model comparison between full and reduced models. Value of $\Delta BIC = BIC(\text{reduced}) - BIC(\text{full})$, for two different reduced models: evoked-only model and diffusion-only model.

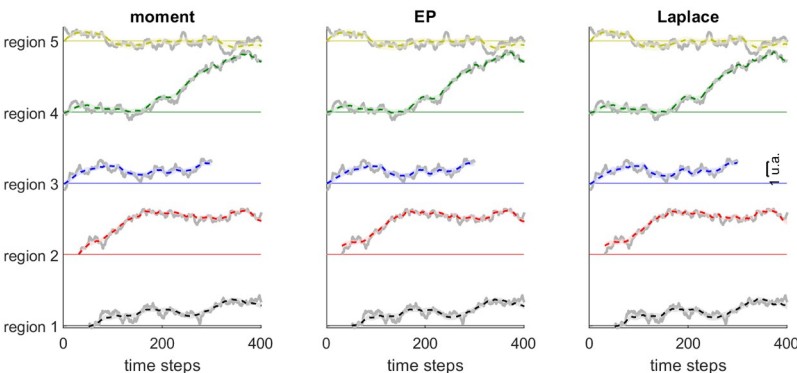

**Fig 3. Trait variables estimated from continuous ETC model, using moment, Laplace or Expectation-Propagation method in the E-step.** The true trait value is represented by thick gray lines, the inferred trait posterior is represented by its posterior mean (dotted coloured lines) and standard deviation (shaded areas). Each colour represent values in a different region. Note that in this example region 2 was created from region 3 at time step 31, region 1 was created de novo at time step 51, while region 3 was suppressed at time step 300.

correctly identified that both horizontal transmission and evoked component contributed to the evolution of the cultural trait (Fig 2c).

## Testing continuous ETC model

Subsequently, we tested the estimation method for ETC models with a continuous trait, and artifacts with count values (the number of fiction books produced in a given region and year). We simulated $K = 5$ regions over 400 times steps (e.g. over 2000 years with time step of $dt = 5$ years; details in Appendix C2 in S1 Appendix). The number of books in the dataset was on average 10 per century and per region, and the values in $A$ correspond in our example to the number of important female characters in each book. We want to reconstruct the dynamics of the support for gender equity (the cultural trait) from this synthetic corpus of fictions. We fitted ETC models from a batch of 100 simulated datasets using the same parameter set, and for each dataset we compared three different methods for approximating the posterior latent distribution during the E-step of the EM algorithm: moment method, Laplace approximation and Expectation-Propagation (EP) (see Appendix B in S1 Appendix).

We found that model fitting based provided very good estimates of true parameters and of the trajectory of cultural traits (Figs 3 and 4f), irrespective of the approximation method used. All three approximation methods also provided good estimates of true parameters, with little to no bias (Fig 4a to 4e). The Laplace and EP approximations provided better estimates than the moment method, with less bias and less variance in parameter estimates. The Laplace and EP approximations provided indeed very similar results, with very high correlation between parameter estimates between the two methods: $r > 0.99$ for all 7 parameters expect noise parameter $\sigma$ ($r = 0.94$). By contrast, the correlation was much lower between estimates given by the moment method and either the EP or Laplace approximation ($r$ in the range 0.2–0.8, except for $\sigma_0^2$ and $x_0$ where $r$ values were in the range 0.94–0.97). We also tested the capacity of the method to detect from the data whether diffusion of the cultural trait between regions and/ or an evoked component are present. For such hypothesis testing, we compared which of four alternative models was better supported by the synthetic data: the full ETC model, that included diffusion and evoked component; the *no diffusion model*, that included evoked component but no diffusion (i.e. parameter $\xi$ is set to 0); the *no evoked model*, that included

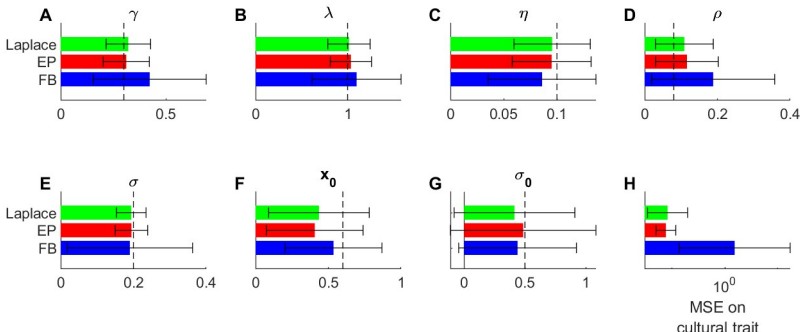

**Fig 4.** A-G Estimated ETC parameters for each type of approximate method in E-step (Laplace approximation, EP or moment method). Dotted line represent the true value. **H**: mean squared error on cultural trait. Bars: average value over datasets; error bars: standard deviation across datasets.

diffusion but no evoked component (i.e. parameter $\lambda$ set to 0); and the *control model*, where diffusion and evoked component are both absent. We compared different metrics to perform model selection: metrics based on penalized log-likelihood (Akaike Information Criterion, AIC; Bayesian Information Criterion, BIC), on cross-validation (Cross-Validated Log-Likelihood, CVLL), or on permutation tests on individual parameters on the full model (boostraps). We report in Fig 5 the proportion of datasets where each of the model is selected, for each metric and each approximation algorithm. Results were similar using either Laplace or EP approximation. For both approximations, the full model was correctly identified in a majority of cases for all metrics. Cross-validation (CVLL) provided the most sensitive measure, as it correctly identified the full model in almost all datasets. Using the moment estimation instead, the full model was correctly identified in a smaller proportion of datasets irrespective of the metrics used. In summary, the EP and Laplace approximations outperformed the moment method for parameter estimation and model selection.

These results were obtained for a certain choice of parameter values and number of observations (i.e. the total number of artifacts). Exploring the estimation properties when all parameter values and number of observations are changed is out of scope of the present study. However, we performed further analyses varying the value of parameter $\lambda$. We found that parameter values were correctly estimated when $\lambda$ was varied (S1 Fig), suggesting that our previous results hold at least for a certain range of parameters. Trait trajectories and model

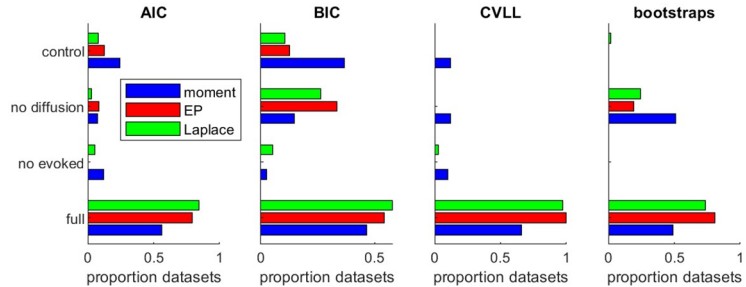

**Fig 5. Model selection for simulated datasets using different metrics (AIC, BIC, CVLL or boostrapping p-values) and approximate inference methods for E-step (moment: Blue bars; Expectation-Propagation: Red bars; Laplace method: Green bars).** Each bar represents the proportion of simulated datasets where the corresponding model (full, no evoked model, no diffusion model or control model) was selected. CVLL was computed using 5-fold cross-validation.

parameters were also correctly estimated when we increased the number of regions to $K = 50$ instead of 5 (S2 Fig). Moreover, in another set of simulations, we varied the number of time steps in each simulation, which effectively modulates the sample size. As expected, we found that when more time steps are included in the dataset, parameters were more accurately estimated (S3 Fig), and that the full model was more often (correctly) identified as the underlying model (S4 Fig). This stresses the importance of sample size for making correct inference from the data, as in any statistical analysis.

Sample size is not the only factor that affects the identifiability of parameters. In Appendix B6 in S1 Appendix, we show that if cultural traits are observed (and not latent variables inferred from cultural artifacts), then parameters ($\gamma$, $\rho$, $\lambda$, $\xi$) can be estimated using simple linear regression. The Variance Inflation Factor (VIF) is a metric that determines the identifiability of parameters in linear regression and thus be applied for each parameter ($\gamma$, $\rho$, $\lambda$, $\xi$). For example, if each region is connected to many regions, then the influence from all neighbouring regions can average out. In such case the parameter $\xi$ cannot be accurately estimated, as its corresponding VIF is very high (S5 Fig, bottom panels). By contrast, the parameter can be estimated accurately in the exact same model if the connectivity between regions is low, as the algorithm can trace back the influence onto a region cultural trait of traits in the few neighbouring regions (S5 Fig, top panels). In general, the cultural traits are not observed directly but we define a pseudo-VIF measure applied *a posteriori* to the estimated model that provides information about the identifiability of each of the parameters that regulate cultural trait evolution (See Appendix B6 in S1 Appendix).

## ETC generates better estimates than a two-stage analysis

Next, we compared our approach to a simpler two-stage approach where cultural traits are first estimated using cultural artifacts produced in the same region and at the same time, and parameters are later estimated directly based on these estimated latents. This second stage resembles the fitting procedure used in [45], applied directed to cultural traits rated by different experts for each region and time. The estimated cultural traits using this method differ from the EM algorithm in two important aspects: they are only constrained by cultural artifacts (whereas in the EM algorithm they are constrained by both artifacts and estimated parameters); they do no include any level of uncertainty. Based on both of these aspects, we can expect poorer estimates: ignoring reliability of the estimates leads to weighing equally data points with many cultural artifacts and data points with a scarcity of artifacts. We compared the two methods on synthetic data generated from an ETC model with 5 regions over 500 time steps, producing continuous-valued artifacts (details in Appendix C3 in S1 Appendix). We separated the time frame into a longer period of scarcity of artifacts (first 400 times steps; mean artifact per time point: 0.1), and a shorter period of abundance of artifacts (last 100 time steps; mean number of artifacts: 1).

Cultural traits estimated from both methods are compared in Fig 6a. We see that the one-stage method provides good estimates of the true cultural traits. It also displays more uncertainty for data points with scarce cultural production (left to the black vertical bar) compared to data points with an abundance of cultural artifacts (right to the black vertical bar). By contrast, the two-stage method provides less smooth and less accurate estimates of cultural traits. We also compare the estimated parameters from both methods in Fig 6b for 30 different simulations. The one-stage method provides very reliable and unbiased estimates, especially for the parameters defining the dynamics of cultural trait evolution ($\gamma$, $\lambda$, $\eta$, $\rho$, $\sigma^2$). Moreover, as predicted, the two-stage method provides very biased and unreliable estimates. We also found that the mean-squared error term over the cultural trait was on average much lower in the

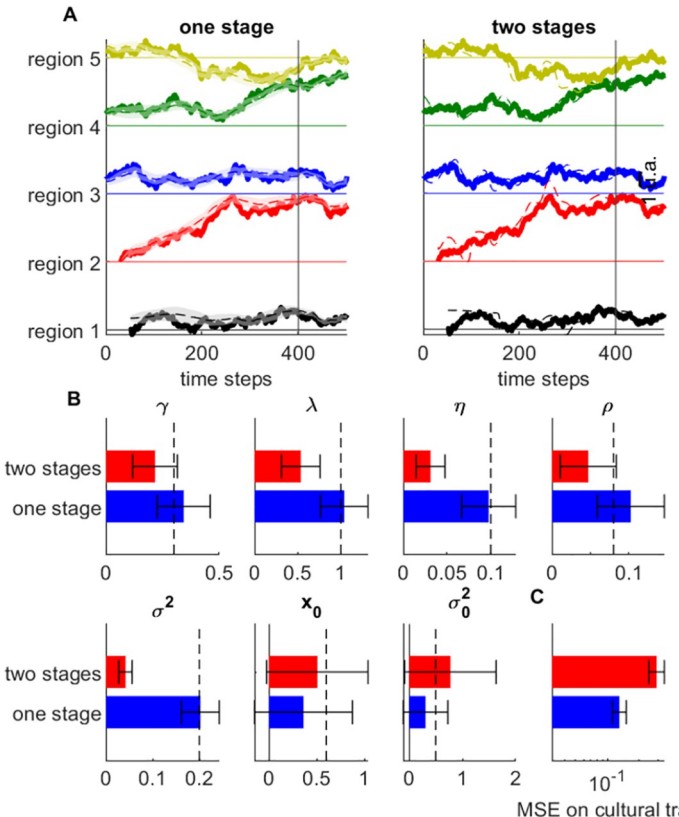

**Fig 6. Comparison of the one-stage and two-stage estimation procedures. A**. Left: Estimates of cultural trait using the standard ETC model. Legend as in Fig 3. Vertical lines marks the transition from low to high cultural production period. Right: Direct estimates of cultural trait by averaging cultural artifacts in same region and within a certain window of time. **B**. Parameter estimates using one-stage (blue bars) vs two-stage (red-bars) method. Vertical dashed line indicate true value of the parameters. Error bars: standard deviation of estimates across datasets. **C**. Average mean-squared-error on cultural traits using one-stage vs. two-stage method (log-scale).

one-stage than in the two-stage procedure (Fig 6c). We attribute this to the fact that the two-stage method estimates the dynamic regime of the cultural trait without taking into account the uncertainty about cultural trait.

## Cultural evolution can also be inferred from synchronous data

Finally, we studied whether ETC model parameters could also be reliably inferred from synchronous instead of diachronous data. We simulated the evolution of a cultural trait in regions forming a phylogenetic tree (details in Appendix C4 in S1 Appendix). The evolution of the cultural trait was influenced by the value of a binary ecological factor (Fig 7a and 7b). We then fitted ETC parameters to the cultural traits observed in all regions at the final time of the simulation. We compared two versions of the model: either the full ETC model, or the model without spatial diffusion, which is very similar to the classical phylogenetic method previously used on cultural datasets [14–16]. Despite using only the final points of the cultural evolution, the method reliably reconstructed the trajectory of the cultural traits along the tree (Fig 7c). The reconstruction error increased when cultural diffusion was not included into the dynamics of cultural evolution (Fig 7d and 7f). Parameters were also estimated relatively well using

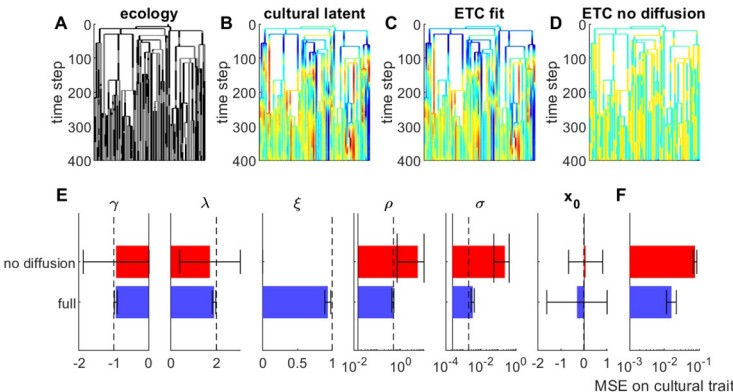

**Fig 7. Fitting ETC model on synchronous dataset. A**. Value of binary ecological factor along phylogenetic tree of regions. The tree represents 50 regions, where parenthood indicates that the new region takes the value of the cultural trait from its parent. Ecological factor is color coded, with blue indicating that the factor is inactive, and red indicating that the factor is active. **B**. Value of cultural trait along the phylogenetic tree, simulated from ETC model. Value is color coded, with blue indicating negative values, and red positive values. **C**. Posterior mean over cultural trait inferred from full ETC model fitted to synchronous data. Same colour code as B. **D**. Same as C, for ETC model that does not include cultural diffusion. **E**. Estimated parameters using full ETC model (blue bars) vs. ETC model without spatial diffusion (red bars). Same legend as Fig 2. **F**. Reconstructing error on cultural trait, for the two variants of the model.

the full ETC model (Fig 7e), with little to no bias. By contrast, ignoring cultural diffusion lead to systematic biases in parameter estimation, especially for the cultural lability parameter $\rho$, which controls vertical transmission. We interpret this as an incorrect assignment of cultural transmission: because horizontal transmission is not taken into account, similarities between the cultural trait in different regions are uniquely attributed to vertical transmission from a common parent.

## Discussion

There has been a recent interest in using quantitative approaches, and in particular model fitting, to test hypotheses about the evolution of culture and cultural traits at a microscopic [46, 47] or macroscopic scale [14, 15, 17]. While phylogenetic approaches have been applied in certain cases [14–17], there are two major differences between the dynamics of cultural and genetic evolution. First, unlike genetic evolution, horizontal transmission is an important factor of cultural evolution, competing with vertical transmission (i.e. transmission from previous generation). Second, unlike Darwinian evolution, cultural evolution may be influenced directly by the environment via phenotypic plasticity [48]. Thus, quantitative models of cultural evolution should capture these three types of driving forces: vertical transmission, horizontal transmission and influence of the environment. We have presented here a new quantitative model termed ETC which captures how these three driving forces can shape the evolution of a cultural trait. In ETC, the importance of these factors are captured by specific parameters. Importantly, we have shown that our model fitting procedure allows to estimate reliably model parameters from a diachronic database, where the evolution of cultural traits are not directly observed but inferred from the production of cultural artificats in different regions and different times. The ETC framework allows to capture different types of cultural traits (as a binary or continuous variable) and cultural artifacts (binary, counts, continuous variable, . . .). At the current time it models the cultural trait as a single metric over the entire population in one region at each time step, but possible extensions could include richer

representations of the cultural traits such as stratified cultural traits by generation [49] or the encoding of various cultural traits [16].

The fitting procedure relies on an Expectation-Maximization algorithm. In the Expectation step, two approximate inference algorithms have shown to yield the best results (for continuous traits): the Laplace approximation and Expectation-Propagation. These approximations work better than the moment method [50] probably because they are global approximations, i.e. they approximate directly the distribution of the collection of trait values at all time steps $T$. By contrast, the moment method relies on local approximations, i.e. approximate the trait value at each time step $T_t$ sequentially, allowing for errors to accumulate from time step to time step. The moment method has a lower computational cost however, and especially it scales better with the number of time steps. The computational complexity of the algorithm is largely driven by inversions of covariance matrices in the E-step. Theses squares matrices are of size $K$ for the moment method, and matrix inversion is repeated at each time step, so the computational cost scales as $O(TK^3)$. By contrast, the GP framework works with the full covariance matrix of size $KT$, so the cost scales as $K^3 T^3$ (the inversion can be performed only for data points with attached observations, so the actual cost can be drastically reduced if the artifacts are sparse). The moment method thus provides a good alternative to the other methods for large datasets.

As expected for any statistical procedure, the accuracy of parameter estimation and model selection depends largely on the structure of the data and in particular on the number of observations (here the number of artifacts; S3–S5 Figs). A full exploration of the estimation error in synthetic data for different values of the parameter set is out of scope of the present study. Rather, prior to using the data on a specific dataset, we encourage to always perform parameter recovery analysis with simulated data matching the original data in all aspects (number of regions, connectivity, value of ecological variables, number of artifacts at each time point) [51]. For example it is expected that for a low number of artifacts the method may not succeed in tearing apart the contributions of horizontal and vertical transmission to cultural alignment. The pseudo-VIF measure (Appendix B6 in S1 Appendix) also provides some information about the identifiability of the parameters *a posteriori*.

We also found that the fitting procedure with EM algorithm is much more reliable than a two-stage procedure where cultural traits are first estimated from cultural artifacts, and ETC parameters are then fitted on these cultural traits (Fig 6). The major reason is that this second method ignores altogether the uncertainty about the cultural trait, which strongly biases the estimates of the dynamic rules that dictate the evolution of these traits. This suggests that fitting and comparing models of cultural evolution based on datasets of cultural traits that ignores the necessary error-in-measure (whether such error comes from limited samples of cultural artifacts or any imprecision in manual ratings of cultural traits) can lead to incorrect conclusions [45]. As a side note, the direct procedure has also three important benefits. First, the uncertainty about the cultural traits naturally reflects the abundance of cultural artifacts in the region (or neighbouring regions) at the time. Second, the temporal and spatial scale over which cultural artifacts are integrated to produce latent trait estimates are not defined a priori but are directly adjusted to the ETC parameters (as implicitly defined from the E step of the EM algorithm). Finally, the dynamics of cultural traits is jointly constrained by cultural artifacts, and the ETC estimated parameters through dynamic rules of the ETC model. This constrasts with the two-step procedure where cultural traits are only inferred from cultural artifacts.

While the method is naturally designed to assess cultural evolution from diachronic datasets, it can also reconstruct the evolution of a cultural trait from a synchronous dataset, i.e. from observing this trait in different regions at a single time (Fig 7). In this sense the method

bears many similarities to existing phylogenetic methods that have been previously employed for cultural datasets. However we found that it is better adapted than phylogenetic methods, as it includes the specificities of cultural evolution with respect to genetic evolution: notably ETC models can incorporate the influence of evoked component and cultural diffusion, unlike standard phylogenetic methods. We found that ignoring horizontal transmission (i.e. cultural diffusion) leads to misattributing shared common traits to vertical transmission (Fig 7). This induces a biased estimation of the cultural lability parameter that controls vertical transmission, as well as unreliable reconstructed evolution of the cultural trait. This suggests a cautious interpretation of previous studies that have used the phylogenetic method to infer co-evolution of cultural traits. For instance, while it is true that Austronesian societies have remained relatively isolated from each other during the last millennium [16], cultural transmission probably occurred many times before the arrival of the Europeans [52]. We believe that taking this horizontal transmission into account could significantly change the way we think cultural traits have evolved in these regions.

The natural application of the method is to study how cultures evolve in different populations or regions, but ETC models could be equally well deployed for quantitative analyses in many other fields. For example, in archaelogy, it could also be used for reconstructing the evolution of cultural traits in a network of connected settlements, where data is accessible to us through fragmented records with usually limited coverage in time and space [53]. Comparison between ETC models with different underlying diffusion networks could be used in this context to infer the pathways of cultural diffusion. In behavioral ecology, ETC models could be applied to understand how cultural traits can diffuse between social groups and whether they can be shaped by ecological factors [54–56]. In studies of human cultures, the different nodes of the network do not need to correspond to geographical regions but could correspond to any kind of social group. For example, ETC models could be used to infer how and why specific cultural traits (e.g. linguistic traits) spread between different social classes, based on a language corpus. The nodes of the network could even be interconnected individuals. For example, ETC models could be used to understand how socioeconomic conditions shape the diffusion and the evolution of artistic practices within a group of artist [57].

## Conclusion

We have proposed a class of model, called Evoked and Transmitted Culture (ETC) models, to capture the dynamics of evolution of cultural traits in a set of interconnected networks. Bayesian methods were shown to provide accurate tool to reconstruct such evolution from a (possibly fragmented) diachronic dataset of cultural artifacts, and also perform well on synchronic datasets. These methods are available as a Matlab toolbox accessible at https://github.com/ahyafil/Evoked_Transmitted_Culture. We hope this new quantitative tool will help unveil the determinants of cultural evolution among societies, social groups and individuals.

## Supporting information

**S1 Appendix.**
(PDF)

**S1 Fig. Estimated parameters of ETC model for different values of the λ parameter.** Each row represents a value of λ, and each column represents one parameter from the ETC model. The last column represents the MSE on the cultural trait. Legend as in Fig 4 of main manuscript.
(PNG)

**S2 Fig. ETC model with $K = 50$ regions.** Panel A shows the true trajectories (full grey lines) and inferred trajectories using the moment method (dotted color lines) for each region. The overlap shows that all trajectories were well estimated. The parameters used for simulations are: $\gamma = 0.1$, $\lambda = 1$, $\xi = 0.1$, $\rho = 0.08$, $\sigma = 0.2$, $x_0 = 0.6$, $\sigma_0 = 2$. Panel B shows the estimated parameters for 100 simulations with same parameter values (bar: mean over simulations; error bar: standard deviation over simulations). Dotted lines indicate the true value of the parameter. (PNG)

**S3 Fig. Estimated parameters of the ETC model with the moment method for different numbers of time steps (100, 400 or 1000 time steps).** The parameters used for simulations are: $\gamma = 0.1$, $\lambda = 1$, $\xi = 0.04$, $\rho = 0.08$, $\sigma = 0.2$, $x_0 = 0.6$, $\sigma_0 = 0.5$. Each panel represents one parameter from the ETC model. Bars and error bars represent the mean and standard deviation of the estimated parameters over 100 simulations, respectively. The bottom right panel represents the MSE on the cultural trait. (PNG)

**S4 Fig. Model comparison for the simulations of S3 Fig.** Each row corresponds to a different number of time steps $T$, each column corresponds to a different metric used for model comparison. The larger is the dataset, the more probable it is that the full model is correctly identified as the best model, for all types of metrics. (PNG)

**S5 Fig. Parameter estimation for ETC models where cultural traits $T$ are directly observed, and for a connectivity matrix $G$ between regions with either low connectivity or high connectivity.** The probability of connectivity (with node strength $G_{rs} = 1$) was set to $p_G = 0.5$ in the high-connectivity model and $p_G = 0.05$ in the low-connectivity model. In each simulation, an ETC model with $K = 50$ regions was simulated during 500 times steps, using the following value of parameters: $\gamma = 0.05$, $\lambda = 1$, $\rho = 0.08$, $\sigma = 0.2$, $x_0 = 0.6$, $\sigma_0 = 0.1$. The value of the diffusion parameter $\xi$ was normalized to the overall probability of connection between each region $p_G$ ($\xi = 0.01/p_G$) so that the overall influence of cultural diffusion was comparable between the high-connectivity and low-connectivity models. **A**. Distribution of parameters estimated from the models, from 500 simulations for both low-connectivity (top row) and high-connectivity (bottom row) matrices. The diffusion parameter is normalized to $\xi/p_G$ for better comparison. Red lines indicate the true value of the parameter. Note that the estimation of parameters $\gamma$, $\xi$ and $\rho$ are much less precise in the high connectivity network compared to the low-connectivity network, while the estimation of $\lambda$ is mostly preserved. **B**. Variance Inflation Factor (VIF) for the regressors related to the $\lambda$, $\xi$ and $\rho$ parameters into the linear regression analysis, averaged over simulations (bar: mean; error bars: standard deviation), for low- and high-connectivity models. VIF values close to 1 indicate good identifiability of the parameters, while values above 5–10 indicate poor identifiability due to regressors being nearly colinear. Note the problem of colinearity in the high-connectiviy network affecting the identifiability of parametres $\xi$ and $\rho$ (but not $\lambda$). **C**. Relationship between VIF for regressor related to $\rho$ and the estimated value of parameter $\rho$. Each dot represents a simulation. Red line indicates the true value. Larger estimation errors are obtained for simulations with large VIF. In other words, the relationship between VIF and estimation error is present within the same set of simulations. (PNG)

## Author Contributions

**Conceptualization:** Alexandre Hyafil, Nicolas Baumard.

**Formal analysis:** Alexandre Hyafil.

**Methodology:** Alexandre Hyafil.

**Software:** Alexandre Hyafil.

**Validation:** Alexandre Hyafil, Nicolas Baumard.

**Writing – original draft:** Alexandre Hyafil, Nicolas Baumard.

**Writing – review & editing:** Alexandre Hyafil, Nicolas Baumard.

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
