## [Decision Letter · Decision Letter 0]

23 Jun 2021

PONE-D-21-13227

Evoked and Transmitted Culture models: Using bayesian methods to infer the evolution of cultural traits in history

PLOS ONE

Dear Dr. HYAFIL,

Thank you for submitting your manuscript to PLOS ONE. After careful consideration, we feel that it has merit but does not fully meet PLOS ONE’s publication criteria as it currently stands. Therefore, we invite you to submit a revised version of the manuscript that addresses the points raised during the review process.

We look forward to receiving your revised manuscript.

Kind regards,

Inés P. Mariño, Ph.D.

Academic Editor

PLOS ONE

Journal Requirements:

Reviewers' comments:

Reviewer's Responses to Questions

**Comments to the Author**

1. Is the manuscript technically sound, and do the data support the conclusions?

Reviewer #1: Yes

Reviewer #2: Yes

2. Has the statistical analysis been performed appropriately and rigorously? 

Reviewer #1: Yes

Reviewer #2: I Don't Know

3. Have the authors made all data underlying the findings in their manuscript fully available?

Reviewer #1: Yes

Reviewer #2: Yes

4. Is the manuscript presented in an intelligible fashion and written in standard English?

Reviewer #1: Yes

Reviewer #2: Yes

5. Review Comments to the Author

Reviewer #1: The authors describe a new method of analyzing diachronous (and synchronous) data with the goal of identifying the effects of evoked and transmitted culture. The authors call their method an Evoked-Transmitted Culture (ETC) model. The paper describes how the model functions, how it can be applied (including calculating uncertainty) and how it compares to some other approaches. Overall, I like this paper a lot and the general idea is clever. However, I think the validation of the method is a little lacking: it is clear this method can work, but I am left very unclear about how reliably it will work and whether there are conditions where it does not do well. I will describe this in more detail below, as well as some more minor points.

Validation

Take, for instance, figure 2 (or any of the other similar figures). These give a very good sense of how the model is working given the values of the parameters assumed in the simulated data set. But what about other parameter values? To really persuade the reader we need a thorough exploration of parameter space to get a sense of how robust this method is. This is particularly important because the resolution of historical data sets might be quite poor compared to rates of cultural adaptation, making it very hard to tell apart an immediate response to the environment or one that accumulates across generations. (I note that the authors acknowledge as much on line 128, although the implications of this for real data sets are not really expanded upon).

Given the number of parameters, such an exploration would be a major endeavor, and so it would be reasonable for the authors to hold off, and save such an exploration for a future paper. But either way, I would encourage them to do so at some point if they want others to take notice of their suggested methodology. Similarly, the authors’ position would be strengthened if they applied their method to real (as opposed to purely simulated) data to show an example of the kind of insights this method offers in practice.

While I do think these are important concerns. I do not think they necessary preclude publication of the paper. This is because what is already there is (as far as I can tell) technically sound, and the paper does make a sensible contribution in its current form. There is simply more that could be done if the authors wish to maximize the impact of their suggested method.

Minor points:

1. lines 13-16. The use of the demographic transition is weird here. Yes, it occurs across societies, but this does not simply imply an ecological response without transmission. For instance, Heidi Colleran has shown the role of social transmission in women’s contraceptive choices in a rural population. While Boyd & Richerson have argued that prestige biased transmission is critical to the demographic transition (see Not By Genes Alone). More generally, there is no unproblematic explanation for the demographic transition as an adaptative ecological response. It remains a big question, that defies a behavioral ecological approach, and cultural transmission seems very important.

2. line 32. Putting “institutions” in the environment and not a part of cultural transmission is a big claim! Many cultural evolutionists would push back against this. If the authors really want to have culturally constructed and transmitted institutions to be part of the environment and not part of transmitted culture then they need to argue why this is the right thing to do in this case.

3. lines 77-89. Overall the model introduction is well written, but it’s a challenge writing it such that even non-statistically oriented scientists can follow. Minimally, I’d unpack the term “probabilistic generative models” just so readers can follow more easily.

4. line 91. It feels odd to introduce phi as a set of parameters, but not explain what they are. Maybe give a couple of examples and then just state this will be explained in full later.

5. Figure 1, I would suggest using different shaped symbols for E, T and A. As it currently is I assumed time flowed from top to bottom, not left to right, so it took me a while to wrap my head around the figure.

6. line 128 - please explain why this is necessary. I think it is because otherwise you might get multiple transitions within a single time step.

7. line 150 - please explain this parameter more. It looks like a constant pull towards 0, proportional to the current distance from 0. Is this right?

8. Equations 8 to 12. I must confess these equations go beyond my mathematical capacity. Hopefully another reviewer can OK them.

9. Figure 3 is really hard to read, the thick lines obscure the shaded areas and thin lines. A different visual approach is required. Also, why do the blue lines stop early? In fact, the x-axis limits are different in all 3 panels - what’s going on?

Reviewer #2: The manuscript Evoked and Transmitted Culture models: Using bayesian methods to infer the evolution of cultural traits in history propose a novel model to infer the evolution of cultural traits. The article is interesting and I support publication should the following points be addressed.

Use Case

The authors decided to introduce the model in rather abstract terms and to test it on synthetic data. This is fine, but it makes it difficult to verify how the model assumptions relate to concrete real-world phenomena, say the evolution of linguistic traits. I was wondering if the authors could present the model with a clear use case instead. This would make it easier to verify to what degree the model assumptions are appropriate for inferring a specific evolutionary process.

The authors explain and motivate the model parameters with many examples (which is good!), but these examples appear to be somewhat unrelated. The example to explain the parameter for artifact productions is the number of universities in the region if artifacts are the number of scientific publications in a certain field, and the example for the data collection process is the number of registered archaeological sites when assessing the fabrication of a particular tool in a region. Again, I would encourage the authors to start with a clear use case in mind, explain all model parameters with respect to the use case and only then show how the model generalizes to other evolutionary stories. It will be much easier for the reader to follow the argument and to verify that the generative model fits the evolutionary process.

Presentation of the model

From what I can tell, the model seems mathematically sound. However the presentation of the model should be improved.

Most importantly, the authors tend to introduce variables and equations without any initial explanation, motivation or justification, line by line adding chunks to the generative story, rather than painting the big picture first and then filling in the details. This makes the manuscript cumbersome to read and the model hard to understand.

A particularly bad example (in terms of didactics) is equation 6 for continuous cultural traits, which throws four previously unmentioned variables at the reader without really providing any explanations on why they are relevant and what they are meant to model.

I collected some other points below:

In equation 1, T_t,r on the left hand side depends on T_t-1, but on the right hand side this conditional event referring to one time stamp earlier has been omitted. Could the authors comment on this? Maybe I do not fully understand the equation.

Theta_T (line 91)

It is rather difficult to follow the argument here, because no further details are given for Theta_T at this point other than the brief explanation that it is a set of parameters of cultural evolution. Theta_T is only explained properly on line 137. Maybe change the order? The same holds true for Theta_A which does not seem to be defined at all.

Equation 4

Could equation 4 be based on relevant theory? What are the implications of this being a linear relationship?

Line 129

Why is this relationship required? Please explain.

Equation 5

Why does equation 4 use a transition rate, but equation 5 the transition rate times delta t?

Equation 6 and 7

Is there any theoretical justification for equation 6 and 7? Could you explain what cultural lability is and why it is relevant here? Why is there a variable to model a potential bias towards positive or negative trait values? Please justify your design decisions.

Line 130: this seems to belong to the simulation, but not the model. Same for line 156.

Both the binary and the cont. model use the same set of parameters theta_T. I would use different parameters here.

Simulation study and scalability

The authors chose to test the model on simulated data, which is fine. I was wondering, though, if the authors could design the simulation with a specific real-world scenario in mind and define the simulation goals accordingly. This could allow others to verify if the scenario is plausible and if all relevant goals are achieved. Moreover, the simulation relies on 6 regions, which seems rather low for real world studies, especially given that the model should be applicable when the nodes of the network are interconnected individuals. How scalable is the approach?

Language

I am not an English native either, but I feel that the language can be improved here and there (especially the use of articles).

Defining ETC models

The paragraph introducing the generative model is rather low on references (line 67 and following). The two definitions for cultural artifacts and cultural traits, for example, appear to be make sense intuitively, but I was wondering if they reflect the scientific consensus. The same holds true for the influencing factors for cultural traits (vertical and horizontal transmission, and ecological factors). I would encourage the authors to provide additional references and introduce the relevant theory to justify the parametrization of the model.

T_t,r and A_t,r(line 87)

Could the authors give a concrete example of a cultural trait T_t,r and a corresponding artifact A_t,r? What would be an existing diachronic data set of cultural artifacts A_t,r? Does a specific A_t,r originate due to a single trait T_t,r, or does it imply the existence of several different traits?

I am lacking the necessary mathematical background to appropriately assess if the general fitting procedure is correct.

6. PLOS authors have the option to publish the peer review history of their article (what does this mean?). If published, this will include your full peer review and any attached files.

Reviewer #1: No

Reviewer #2: No

---

## [Author Response · Author response to Decision Letter 0]

30 Dec 2021

We thank both referees for their in-depth reviews of our manuscript. We have addressed their comments – we think that the paper has now improved both in quality and clarity, and hope it is suitable for publication in Plos One.

Reviewer #1: The authors describe a new method of analyzing diachronous (and synchronous) data with the goal of identifying the effects of evoked and transmitted culture. The authors call their method an Evoked-Transmitted Culture (ETC) model. The paper describes how the model functions, how it can be applied (including calculating uncertainty) and how it compares to some other approaches. Overall, I like this paper a lot and the general idea is clever. However, I think the validation of the method is a little lacking: it is clear this method can work, but I am left very unclear about how reliably it will work and whether there are conditions where it does not do well. I will describe this in more detail below, as well as some more minor points.

Validation

Take, for instance, figure 2 (or any of the other similar figures). These give a very good sense of how the model is working given the values of the parameters assumed in the simulated data set. But what about other parameter values? To really persuade the reader we need a thorough exploration of parameter space to get a sense of how robust this method is. This is particularly important because the resolution of historical data sets might be quite poor compared to rates of cultural adaptation, making it very hard to tell apart an immediate response to the environment or one that accumulates across generations. (I note that the authors acknowledge as much on line 128, although the implications of this for real data sets are not really expanded upon).

Given the number of parameters, such an exploration would be a major endeavor, and so it would be reasonable for the authors to hold off, and save such an exploration for a future paper. But either way, I would encourage them to do so at some point if they want others to take notice of their suggested methodology.

The reviewer raises an important point. As noted by the reviewer, the parameter space of the model is simply too large to allow for a thorough exploration of the estimation properties throughout this parameter space. As a first step towards this endeavor, we have replicated the estimation for the continuous ETC model, varying the values of parameter λ (the influence of the ecological variable) and the number of regions (50 regions instead of 5). The good estimation properties of the algorithm were maintained for all values of λ, and also for a larger number of regions. In other simulations, we varied the number of time steps in the simulation, which is equivalent to changing the sample size. As expected, the more time steps are included in the simulations, the more accurate are parameter estimates, and the more often is the correct model identified using model selection tools. We have also added a new tool (the pseudo Variance Inflation Factor) to estimate the identifiability of parameters of cultural trait evolution for continuous traits (see new Section B.6). We introduce the tool in Results section and illustrate it with the case of a large number of connected regions that result in the poor identifiability of the cultural diffusion parameter. The new results are presented as Supplementary Figures 1-5, with the following addition in main text:

“These results were obtained for a certain choice of parameter values and number of observations (i.e. the total number of artifacts). Exploring the estimation properties when all parameter values and number of observations are changed is out of scope of the present study. However, we performed further analyses varying the value of parameter λ. We found that parameter values were correctly estimated when λ was varied (Supplementary Figure 1), suggesting that our previous analysis hold at least for a certain range of parameters. Trait trajectories and model parameters were also correctly estimated when we increased the number of regions to K=50 instead of 5 (Supplementary Figure 2). Moreover, in another set of simulations, we varied the number of time steps in each simulation, which effectively modulates the sample size. As expected, we found that when the number of time steps was larger, parameters were more accurately estimated (Supplementary Figure 3), and that the full model was more often (correctly) identified as the underlying model (Supplementary Figure 4). This stresses the importance of sample size for making correct inference from the data, as in any statistical analysis.

Sample size is not the only factor that affects the identifiability of parameters. In section B.6, we show that if cultural traits are observed (and not latent variables inferred from cultural artifacts), then parameters (γ,ρ,λ,ξ) can be estimated using simple linear regression. The Variance Inflation Factor (VIF) is a metric that determines the identifiability of parameters in linear regression and thus be applied for each parameter (γ,ρ,λ,ξ) . For example, if each region is connected to many regions, then the influence from all neighbouring regions can average out. In such case the parameter ξ cannot be accurately estimated, as its corresponding VIF is very high (Supp. Figure 5, bottom panels). By contrast, the parameter can be estimated accurately in the exact same model if the connectivity between regions is low, as the algorithm can trace back the influence onto a region cultural trait of traits in the few neighbouring regions (Supp. Figure 5, top panels). In general, the cultural traits are not observed directly but we define a pseudo-VIF measure applied a posteriori to the estimated model that provides information about the identifiability of each of the parameters that regulate cultural trait evolution (See Section B.6).”

Overall, as for any statistical modelling technique, the best approach is to validate the estimation procedure before applying it to experimental data on synthetic data that matches as much as possible the experimental data (same number of regions, artifacts, credible values of parameters). This can be done in a straightforward manner by any researcher interested in using our method for their dataset by some simple adaptation of the code. We have now added in the Discussion:

“As expected for any statistical procedure, the accuracy of parameter estimation and model selection depends largely on the structure of the data and in particular on the number of observations (here the number of artifacts). A full exploration of the estimation error in synthetic data for different values of the parameter set is out of scope of the present study. Rather, prior to using the data on a specific dataset, we encourage to always perform parameter recovery analysis with simulated data matching the original data in all aspects (number of regions, connectivity, value of ecological variables, number of artifacts at each time point) (Wilson et al., 2019). For example, it is expected that for a low number of artifacts the method may not succeed in tearing apart the contributions of horizontal and vertical transmission to cultural alignment. The pseudo-VIF measure (Section B.6) also provides some information about the identifiability of the parameters a posteriori.”

Similarly, the authors’ position would be strengthened if they applied their method to real (as opposed to purely simulated) data to show an example of the kind of insights this method offers in practice.

We have actually applied the methods to a dataset of literary works across ages and regions of the world, to understand how the concept of romanticism has evolved and its relation to the affluence of the population. This analysis is part of a broader set of analyses to test different hypotheses regarding the cultural evolution of romanticism. We feel this practical example cannot be presented here in a compact fashion, and will point to the other manuscript (soon to be published) for a case example of the method.

While I do think these are important concerns. I do not think they necessary preclude publication of the paper. This is because what is already there is (as far as I can tell) technically sound, and the paper does make a sensible contribution in its current form. There is simply more that could be done if the authors wish to maximize the impact of their suggested method.

Minor points:

1. lines 13-16. The use of the demographic transition is weird here. Yes, it occurs across societies, but this does not simply imply an ecological response without transmission. For instance, Heidi Colleran has shown the role of social transmission in women’s contraceptive choices in a rural population. While Boyd & Richerson have argued that prestige biased transmission is critical to the demographic transition (see Not By Genes Alone). More generally, there is no unproblematic explanation for the demographic transition as an adaptative ecological response. It remains a big question, that defies a behavioral ecological approach, and cultural transmission seems very important.

We understand the point of view of the reviewer, and we agree that "there is no unproblematic explanation for the demographic transition as an adaptative ecological response". However, there are many adaptive approaches to the demographic transition (some arguing that it is adaptive, some arguing that it is a mismatch). In fact, Colleran publishes her article in a special issue of Philosophical Transaction ‘Understanding variation in human fertility: what can we learn from evolutionary demography?' in which most articles discuss adaptive responses to ecological changes. Similarly, in economics, a popular explanation is that the demographic transition is driven by the increase in access to education, which is an environmental change (i.e. the relative cost of education decreases), and does not necessarily requires cultural transmission. In this perspective, such an example does not seem weird at all.

We have nuanced the sentence:

"A typical example here is the number of children per woman (Sear et al 2016; Shen et al., 2016; Lawson et al 2016): the environment in which this individual lives (in terms of income, social support, interpersonal violence) is an important indicator of an individual’s reproductive preference (although transmitted norms also probably play a role (Colleran, 2016))."

Note also that the following paragraph opens with another nuance:

"Obviously, transmitted and evoked cultures are located on the same continuum. All evoked cultural behaviors are also partially influenced by cultural transmission. "

Colleran, H. (2016). The cultural evolution of fertility decline. Philosophical Transactions of the Royal Society B: Biological Sciences, 371(1692), 20150152.

To better ground this example in the literature in cultural evolution, we now cite the following articles: 

Sear, R., Lawson, D. W., Kaplan, H., & Shenk, M. K. (2016). Understanding variation in human fertility: what can we learn from evolutionary demography?. Philosophical Transactions of the Royal Society B: Biological Sciences, 371(1692), 20150144.

Shenk, M. K., Kaplan, H. S., & Hooper, P. L. (2016). Status competition, inequality, and fertility: implications for the demographic transition. Philosophical Transactions of the Royal Society B: Biological Sciences, 371(1692), 20150150.

Lawson, D. W., & Borgerhoff Mulder, M. (2016). The offspring quantity–quality trade-off and human fertility variation. Philosophical Transactions of the Royal Society B: Biological Sciences, 371(1692), 20150145.

2. line 32. Putting “institutions” in the environment and not a part of cultural transmission is a big claim! Many cultural evolutionists would push back against this. If the authors really want to have culturally constructed and transmitted institutions to be part of the environment and not part of transmitted culture then they need to argue why this is the right thing to do in this case.

We agree that this is not the right place to have a debate as to whether institutions are information inheritance or environment inheritance. We have deleted the term. 

3. lines 77-89. Overall the model introduction is well written, but it’s a challenge writing it such that even non-statistically oriented scientists can follow. Minimally, I’d unpack the term “probabilistic generative models” just so readers can follow more easily.

We agree with the referee that it is a real challenge. Following the reviewer’s advice, we have now unpacked:

“A probabilistic generative model refers to a parametric model which defines a probability distribution for the observations (here, the cultural artifacts) when the input variables of the models (here, the number of regions and ecological variables) are known. Bayesian methods allow notably to estimate the parameters of the generative model that best fit the observations in the dataset (e.g. through Maximum Likelihood Estimation) and compare which of a set of models is best supported by the observations”.

4. line 91. It feels odd to introduce phi as a set of parameters, but not explain what they are. Maybe give a couple of examples and then just state this will be explained in full later.

Thank you for the suggestion. We now detail:

“θT represents a set of parameters of cultural evolution (which control notably the level of horizontal transmission, vertical transmission and evoked component, see details in following sections)”

5. Figure 1, I would suggest using different shaped symbols for E, T and A. As it currently is I assumed time flowed from top to bottom, not left to right, so it took me a while to wrap my head around the figure.

We followed the referee’s advice and now use different shapes for E, T and A.

6. line 128 - please explain why this is necessary. I think it is because otherwise you might get multiple transitions within a single time step.

The reviewer’s intuition is correct. We now make it explicit:

“If the time step is too large, then in equations [3] the probability of transition in one time step is not infinitesimal, so the transition rates Rup(r,t) and Rdown(r,t) cannot be approximated by a constant during each time step due to their dependence in cultural traits (see below).”

7. line 150 - please explain this parameter more. It looks like a constant pull towards 0, proportional to the current distance from 0. Is this right?

This is correct. This is a standard term in dynamical systems that parametrizes how fast the memory of a state is lost. We now detail:

“ρ determines the time scale at which the memory of cultural traits is lost. For example, in an isolated region (no neighbour), in the absence of ecological variable and noise, then Ttr converges exponentially to γ /ρ at time scale 1/ρ, i.e. the cultural trait is "forgotten" with a time scale of 1/ρ”

8. Equations 8 to 12. I must confess these equations go beyond my mathematical capacity. Hopefully another reviewer can OK them.

Unfortunately, it does not seem that the other reviewer can either. We hope the good performance of the estimation procedure argues in favor of their validity.

9. Figure 3 is really hard to read, the thick lines obscure the shaded areas and thin lines. A different visual approach is required. 

We agree with the reviewer. We have changed the colour and width of the difference curves, which in our opinion makes it easier to compare the true and estimated trajectories of the cultural trait.

Also, why do the blue lines stop early?

This is because the corresponding region is suppressed at the corresponding time. We illustrated here the possibility that some states are created, divided or suppressed at certain times. This was described in the corresponding Appendix C2, but we now make it explicit in the legend of the figure:

“Note that in this example region 2 was created from region 3 at time step 31, region 1 was created de novo at time step 51, while region 3 was suppressed at time step 300.”

 In fact, the x-axis limits are different in all 3 panels - what’s going on?

This was an unfortunate error. It has been corrected. 

Reviewer #2: The manuscript Evoked and Transmitted Culture models: Using bayesian methods to infer the evolution of cultural traits in history propose a novel model to infer the evolution of cultural traits. The article is interesting and I support publication should the following points be addressed.

Use Case

The authors decided to introduce the model in rather abstract terms and to test it on synthetic data. This is fine, but it makes it difficult to verify how the model assumptions relate to concrete real-world phenomena, say the evolution of linguistic traits. I was wondering if the authors could present the model with a clear use case instead. This would make it easier to verify to what degree the model assumptions are appropriate for inferring a specific evolutionary process.

We thank the reviewer for raising the important issue of clarity. Following the reviewer advice, we now present a what the different variables mean for a possible example. The presentation of the introduction now reads:

“We will use the following example as an illustration: in this example, the cultural trait is the general support for gender equity in the population (a cognitive disposition), while the cultural artifacts could be a corpus of fictions, and for each fiction in the corpus we measure (quantitatively) the importance of female characters. We expect that the cognitive disposition at a given time and region will influence the content of the artifacts: the more support there is for gender equity in the population, the more likely we expect to find important female characters in the fictions produced at that time and region.

A general question in cultural evolution is to understand the relative importance of ecological factors, vertical transmission and horizontal transmission in the evolution of the cultural trait. Because we have no direct access to the mental dispositions of ancient populations, we want to infer the dynamics of the cultural traits using the indirect observations of cultural artifacts. In our example, we would want to understand what drives changes in the support for gender equity in the population from a dataset of fiction books at different times and regions, each classified according to the importance of its female characters. If our hypothesis is that economic development leads to higher support of gender equity, we could use the GDP per habitant (a proxy to economic development) as the ecological factor. (…) Bayesian methods allow notably to estimate the parameters of the generative model that best fit the observations in the dataset (e.g. through Maximum Likelihood Estimation) and compare which of a set of models is best supported by the observations. In our example, we could compare a model where economic development is taken as the ecological factor driving changes in support for gender equity with a model where the average level of science education is the ecological factor.

(…)

In the following sections we discuss two distinct classes of ETC models: when the cultural trait takes the form of a binary value and when it takes the form of a continuous value. In our example, these two classes correspond to classifying a population as supporting or non-supporting gender equity for the binary trait; or to quantifying the support to gender equity in the population with a continuous value.”

The authors explain and motivate the model parameters with many examples (which is good!), but these examples appear to be somewhat unrelated. The example to explain the parameter for artifact productions is the number of universities in the region if artifacts are the number of scientific publications in a certain field, and the example for the data collection process is the number of registered archaeological sites when assessing the fabrication of a particular tool in a region. Again, I would encourage the authors to start with a clear use case in mind, explain all model parameters with respect to the use case and only then show how the model generalizes to other evolutionary stories. It will be much easier for the reader to follow the argument and to verify that the generative model fits the evolutionary process.

We have also changed the examples related to artifact productions to be in line with the single example provided above:

“The production of artifacts is guided by the value of the cultural traits, and possibly as well by some other (known) factors Frt that can relate to the material conditions of artifact productions in region r at time t (e.g. the number of fiction books published) and to the data collection process (e.g. the records of ancient publications)”

Presentation of the model

From what I can tell, the model seems mathematically sound. However the presentation of the model should be improved.

Most importantly, the authors tend to introduce variables and equations without any initial explanation, motivation or justification, line by line adding chunks to the generative story, rather than painting the big picture first and then filling in the details. This makes the manuscript cumbersome to read and the model hard to understand.

A particularly bad example (in terms of didactics) is equation 6 for continuous cultural traits, which throws four previously unmentioned variables at the reader without really providing any explanations on why they are relevant and what they are meant to model.

Equation 6 is central to the model so the reviewer is right to ask for more clarity. We have now subsequently developed the explanation of terms in the equation:

“Parameter ρ is the cultural lability (or cultural leak). It determines the time scale at which the memory of cultural traits is lost. For example, in an isolated region (no neighbour), in the absence of ecological variable and noise, then Ttr converges exponentially to γ/ρ at time scale 1/ρ, i.e. the cultural trait is "forgotten" with a time scale of 1/ρ. γ is a bias towards positive or negative trait values capturing a "default" tendency of the cultural trait in the absence of evoked component and horizontal transmission. η is a Wiener capturing the stochastic component in cultural evolution, and σ is the noise parameter. As for the binary case, λ represents susceptibility to the external factor, and ξ the rate of cultural diffusion. Equation 6 can be viewed as an Ornstein-Uhlenbeck process where the drift term depends linearly on the value of the ecological variable and cultural traits in neighbouring regions, while the diffusion term is constant.”

We have tried to clarify better the choice of variables and equations throughout the presentation of the model (see tracked changes). We hope this will make it more digestible now.

I collected some other points below:

In equation 1, T_t,r on the left hand side depends on T_t-1, but on the right hand side this conditional event referring to one time stamp earlier has been omitted. Could the authors comment on this? Maybe I do not fully understand the equation.

We thank the reviewer for noting this error on the equation. It has been corrected.

Theta_T (line 91)

It is rather difficult to follow the argument here, because no further details are given for Theta_T at this point other than the brief explanation that it is a set of parameters of cultural evolution. Theta_T is only explained properly on line 137. Maybe change the order? The same holds true for Theta_A which does not seem to be defined at all.

We thank the reviewer for the suggestion we now detail:

“Theta_T represents a set of parameters of cultural evolution (which control notably the level of horizontal transmission, vertical transmission and evoked component, see details in following sections)”

“For example, if each cultural artifact is a binary variable, then we could define p(Atr) = σ(β0 + β1 Ttr + β2 Ftr) where σ is the logistic function and θA = { β0, β1, β2} are parameters that control the bias and the influence of the trait and factor on the cultural artifact, respectively.”

Equation 4

Could equation 4 be based on relevant theory? What are the implications of this being a linear relationship?

We main reasons for choosing a linear relationship are simplicity of the model, interpretability (each parameter characterizes the influence of one variable onto the transition rates), and simplifying the fitting algorithm (most notably the M-step of the EM equation). More complex frameworks have been developed for controlling transitions in a discrete Markov chain, e.g. Bengio & Frasconi 1995, but at the cost of interpretability. Actually the linear assumption is equivalent to the model of Bengio & Frasconi with a single layer network defining the transition rates from the variables. We have now added:

“For the sake of simplicity and interpretability of the model, we propose here linear effects, i.e. the transitions rates depend linearly on the value of the ecological factor and cultural traits in neighbouring regions”

Line 129

Why is this relationship required? Please explain.

We now explain:

“If the time step is too large, then in equations [3] the probability of transition in one time step is not infinitesimal, so the transition rates Rup(r,t) and Rdown(r,t) cannot be approximated by a constant during each time step due to their dependence in cultural traits (see below).”

Equation 5

Why does equation 4 use a transition rate, but equation 5 the transition rate times delta t?

We thank the reviewer for noting the discrepancy. The dt term was incorrect in equation 5, is has been removed.

Equation 6 and 7

Is there any theoretical justification for equation 6 and 7? 

Equation 6 can be viewed as an Ornstein-Uhlenbeck process where the drift term depends linearly on the value of the ecological variable and cultural traits in neighbouring regions and the diffusion term is constant. The Orstein-Uhlenbeck is the standard stochastic process used to describe a variable that evolves through time with a certain drift, noise and default value (the strength of the force towards the default value, parametrized by parameter ρ, determines the speed at which a cultural trait can be lost, i.e. vertical transmission). Equation 7 corresponds to the discretization of Equation 6 (required as all datasets rely on discrete times). We have added the following sentence in the manuscript:

“Equation 6 be viewed as an Ornstein-Uhlenbeck process where the drift term depends linearly on the value of the ecological variable and cultural traits in neighbouring regions, while the diffusion term is constant.”

Could you explain what cultural lability is and why it is relevant here? Why is there a variable to model a potential bias towards positive or negative trait values? Please justify your design decisions.

We agree with the reviewer that these terms deserved a better introduction. We have now expanded the explanation:

“ρ determines the time scale at which the memory of cultural traits is lost. For example, in an isolated region (no neighbour), in the absence of ecological variable and noise, then Ttr converges exponentially to γ /ρ at time scale 1/ρ, i.e. the cultural trait is "forgotten" with a time scale of 1/ρ. γ is a bias towards positive or negative trait values capturing a "default" tendency of the cultural trait in the absence of evoked component and horizontal transmission.”

Line 130: this seems to belong to the simulation, but not the model. Same for line 156.

The initial value (or initial distribution) of the latent factor (i.e. cultural trait) are also parts of the definition of a latent probabilistic model, usually with their own set of parameters. This is the case here: x0 (and σ0 in the case of cultural traits) are parameters from the cultural evolution model that are estimated from the dataset. We now detail:

“Initial values are parametrized by p(T1r=1)=x0 (and also for regions created de novo), with x0 the probability that the cultural trait is present in any new region.”

Both the binary and the cont. model use the same set of parameters theta_T. I would use different parameters here.

We disagree with the reviewer here. Following the standards in the statistics / machine learning literature, we define a vector θT that represents the parameter set for cultural evolution for any model of cultural trait evolution, and then details what this parameter set is composed of for different subclasses of models. We use the same name for parameters in θT when their interpretation is similar (i.e. λ is the sensitivity to the ecological factor, ξ controls the level of horizontal transmission, x0 sets the initial value), while other parameters are specific to binary or continuous cultural traits.

Simulation study and scalability

The authors chose to test the model on simulated data, which is fine. I was wondering, though, if the authors could design the simulation with a specific real-world scenario in mind and define the simulation goals accordingly. This could allow others to verify if the scenario is plausible and if all relevant goals are achieved. 

We have added the following in the presentation of the model with a continuous latent variable:

“Subsequently, we tested the estimation method for ETC models with a continuous trait, and artifacts with count values (the number of fiction books produced in a given region and year). We simulated K=5 regions over 400 times steps (e.g. over 2000 years with time step of dt=5 years). The number of books in the dataset was on average 10 per century and per region, and the values in A correspond in our example to the number of important female characters in each book. We want to reconstruct the dynamics of the support for gender equity (the cultural trait) from this synthetic corpus of fictions.”

Moreover, the simulation relies on 6 regions, which seems rather low for real world studies, especially given that the model should be applicable when the nodes of the network are interconnected individuals. How scalable is the approach?

We thank the reviewer for raising this important point. The scalability differs between the case of binary vs continuous traits. The different methods in the continuous trait model scale cubically with the number of regions K, so they can be applied as long as K remains somehow lower than 80-100. For large number of regions, the moment method should be favored as it scales linearly with the number of time steps so the overall computational cost is not prohibitive. We now illustrate the method on such large number of regions: we simulated and fitted the model using the moment method with now K=50. The results are presented in the new Supplementary Figure 2: the trajectories of cultural traits and the model parameters are well estimated. This figure is presented in main text:

“Trait trajectories and model parameters were also correctly estimated when we increase the number of regions to K=50 instead of 5 (Supplementary Figure 2).”

We also added the following in the discussion:

“The computational complexity of the algorithm is largely driven by inversions of covariance matrices in the E-step. Theses squares matrices are of size K for the moment method, and matrix inversion is repeated at each time step, so the computational cost scales as O(TK3). By contrast, the GP framework works with the full covariance matrix of size KT, so the cost scales as K3T3 (the inversion can be performed only for data points with attached observations, so the actual cost can be drastically reduced if the artifacts are sparse).”

The method however does not scale well for binary traits as we explicitly represent all combinations of cultural trait values (i.e. 2K combinations). We moved one sentence from the Annexes to the main text to make this point more explicit:

“For large number of regions K (more than 10-20), the number of combinations becomes prohibitive, so we must resort to approximate solutions using variational inference or sampling methods (Gharamani et al, 1997).”

Language

I am not an English native either, but I feel that the language can be improved here and there (especially the use of articles).

We have tried to improve the English.

Defining ETC models

The paragraph introducing the generative model is rather low on references (line 67 and following). The two definitions for cultural artifacts and cultural traits, for example, appear to be make sense intuitively, but I was wondering if they reflect the scientific consensus. The same holds true for the influencing factors for cultural traits (vertical and horizontal transmission, and ecological factors). I would encourage the authors to provide additional references and introduce the relevant theory to justify the parametrization of the model.

The distinction between vertical and horizontal transmission, and ecological factors is standard. See for instance the three papers we cite (all highly cited) and the following Wikipedia article: https://en.wikipedia.org/wiki/Evolutionary_psychology_and_culture#Evoked_and_transmitted_culture:

Nettle, D. (2009). Beyond nature versus culture: cultural variation as an evolved characteristic. Journal of the Royal Anthropological Institute, 15(2), 223-240.

Gangestad, S. W., Haselton, M. G., & Buss, D. M. (2006). Toward an Integrative Understanding of Evoked and Transmitted Culture: The Importance of Specialized Psychological Design. Psychological Inquiry, 17(2), 138-151.

Tooby, J., & Cosmides, L. (1992). The psychological foundations of culture. The adapted mind: Evolutionary psychology and the generation of culture, 19.

We have also added following citation:

Scott-Phillips, T. C., Dickins, T. E., & West, S. A. (2011). Evolutionary theory and the ultimate–proximate distinction in the human behavioral sciences. Perspectives on Psychological Science, 6(1), 38-47.

Regarding the distinction between cultural artifacts and cultural traits, we now better explain what we mean (additions are in italics):

"We first distinguish between cultural traits and cultural artifacts. Cultural artifacts correspond to the material production of a given population: tools, songs, portraits, watches, novels, etc. (see Figure 1). By contrast, cultural traits are the cognitive and behavioral dispositions shared by the member of the population: social trust, belief in a moralizing god, conservativeness, etc. Recent developments in cognitive sciences suggest that it is possible to infer cultural traits from cultural artifacts. For instance, action facial unites in portraits provide information about the social traits (trustworthiness, dominance) sitters want to display (Oosterhof & Todorov, 2008); the minor mode, a set of tones that is generally associated with subdued, sad or dark emotions (Bowling et al., 2012) and enjoyed preferably by individuals high on empathy (Kawakami & Katahira, 2015; Taruffi & Koelsch, 2014), provides information about the personality traits of the audience; imaginary worlds in fictions is associated with higher openness-to-exploration and preferences for explorations (Nave et al. ; Dubourg and Baumard, 2021).

In line with these developments, recent works have used a diversity of cultural artefacts such as portraits (Safra et al., 2020), theater plays (Martins & Baumard, 2020), music (Benetos et al, 2021) and movies (Dubourg et al., 2021) to infer the evolution of cultural traits such as social trust (Safra et al., 2020; Martins & Baumard, 2020), positive and negative moods (Acerbi et al., 2013), wellbeing (Hills et al., 2019; Benetos et al., 2021), individualism (Twenge et al. 2012; Yu et al. 2016), romantic love (Martins & Baumard, 2022) and exploratory preferences (Dubourg et al., 2021). In all these cases, scientists build on cognitive and behavioral sciences to connect a specific cultural aspect of artefact (the smile in portrait) with the underlying cultural trait (the priority given to appearing trustworthy), and then reconstruct the long-term evolution of this cultural trait using a long-term series of cultural artefacts.

Cultural traits can themselves be influenced by several factors: they can be transmitted from generation to generation (vertical cultural transmission), transmitted from one society to its neighbor (horizontal cultural transmission) or evoked by ecological factors (socio-economic development, political organization, etc.).”

Bowling, D. L., Sundararajan, J., Han, S., & Purves, D. (2012). Expression of emotion in Eastern and Western music mirrors vocalization. PLoS One, 7(3), e31942.

Kawakami, A., & Katahira, K. (2015). Influence of trait empathy on the emotion evoked by sad music and on the preference for it. Frontiers in Psychology, 6. https://doi.org/10.3389/fpsyg.2015.01541

Dubourg, E., & Baumard, N. (2021). Why Imaginary Worlds?: The psychological foundations and cultural evolution of fictions with imaginary worlds. Behavioral and Brain Sciences, 1-52.

Nave, G., Rentfrow, J., & Bhatia, S. (2020). We are what we watch: Movie plots predict the personalities of those who “like” them.

Oosterhof NN, Alexander Todorov. The functional basis of face evaluation

Proceedings of the National Academy of Sciences Aug 2008, 105 (32) 11087-11092; DOI: 10.1073/pnas.0805664105 

Acerbi, A., Lampos, V., Garnett, P., & Bentley, R. A. (2013). The expression of emotions in 20th century books. PloS one, 8(3), e59030.

Hills, C. Illushka Seresinhe, E. Proto, D. Sgroi, Historical Analysis of National Subjective Wellbeing using Millions of Digitized Books. Nat. Hum. Behav. (2019).

Twenge, J. M., Campbell, W. K., & Gentile, B. (2012). Increases in individualistic words and phrases in American books, 1960–2008. PloS one, 7(7), e40181.

Yu, F., Peng, T., Peng, K., Tang, S., Chen, C. S., Qian, X., ... & Chai, F. (2016). Cultural value shifting in pronoun use. Journal of Cross-Cultural Psychology, 47(2), 310-316.

Benetos, E., Ragano, A., Sgroi, D., & Tuckwell, A. (2021). Measuring national happiness with music. University of Warwick, Department of Economics.

Dubourg, E., Thouzeau, V., de Dampierre, C., & Baumard, N. (2021). Exploratory preferences explain the cultural success of imaginary worlds in modern societies

Safra, L., Chevallier, C., Grèzes, J., & Baumard, N. (2020). Tracking historical changes in trustworthiness using machine learning analyses of facial cues in paintings. Nature communications, 11(1), 1-7

Martins, M. D. J. D., & Baumard, N. (2020). The rise of prosociality in fiction preceded democratic revolutions in Early Modern Europe. Proceedings of the National Academy of Sciences, 117(46), 28684-28691.

T_t,r and A_t,r(line 87)

Could the authors give a concrete example of a cultural trait T_t,r and a corresponding artifact A_t,r? What would be an existing diachronic data set of cultural artifacts A_t,r? 

We believe we have addressed all these points with the example used in the manuscript where the cultural trait would be the support for gender equity and cultural artifacts would be a database of fictions (produced at different regions and times) where the importance of female characters is evaluated. 

Does a specific A_t,r originate due to a single trait T_t,r, or does it imply the existence of several different traits?

As explained in the Discussion:

“At the current time it models the cultural trait as a single metric over the entire population in one region at each time step, but possible extensions could include richer representations of the cultural traits such as stratified cultural traits by generation (Fogarty et al., 2019) or the encoding of various cultural traits (Watts et al, 2016).”

I am lacking the necessary mathematical background to appropriately assess if the general fitting procedure is correct.

---

## [Decision Letter · Decision Letter 1]

27 Jan 2022

PONE-D-21-13227R1Evoked and Transmitted Culture models: Using bayesian methods to infer the evolution of cultural traits in historyPLOS ONE

Dear Dr. HYAFIL,

Thank you for submitting your manuscript to PLOS ONE. After careful consideration, we feel that it has merit but does not fully meet PLOS ONE’s publication criteria as it currently stands. Therefore, we invite you to submit a revised version of the manuscript that addresses the points raised during the review process. The reviewers have been very positive about the revised version that you submitted and there are only a few minor issues that need to be addressed.

We look forward to receiving your revised manuscript.

Kind regards,

Inés P. Mariño, Ph.D.

Academic Editor

PLOS ONE

Journal Requirements:

Reviewers' comments:

Reviewer's Responses to Questions

**Comments to the Author**

1. If the authors have adequately addressed your comments raised in a previous round of review and you feel that this manuscript is now acceptable for publication, you may indicate that here to bypass the “Comments to the Author” section, enter your conflict of interest statement in the “Confidential to Editor” section, and submit your "Accept" recommendation.

Reviewer #1: All comments have been addressed

Reviewer #2: All comments have been addressed

2. Is the manuscript technically sound, and do the data support the conclusions?

Reviewer #1: Yes

Reviewer #2: Yes

3. Has the statistical analysis been performed appropriately and rigorously? 

Reviewer #1: Yes

Reviewer #2: Yes

4. Have the authors made all data underlying the findings in their manuscript fully available?

Reviewer #1: Yes

Reviewer #2: Yes

5. Is the manuscript presented in an intelligible fashion and written in standard English?

Reviewer #1: Yes

Reviewer #2: Yes

6. Review Comments to the Author

Reviewer #1: The authors have revised and resubmitted their manuscript describing ETC models to evaluate the effects of cultural transmission and environmental factors on cultural change. I liked the paper on its initial submission and find it to be considerably improved following the authors’ revisions. I thank the authors for their work following the previous round of review, and I now consider the manuscript suitable for publication, although I will note two minor points:

line 32 – I appreciate the authors removal of the word institutions here, but there remains an issue with wording. As it stands, the authors now seem to be putting various features of the environment within “culture”, for instance pathogen presence. However, this is not typical. For instance, environmental features that arose independently of human actions might nonetheless influence how we behave, yet to include these features themselves as “culture” is unusual because it is not clear how they are being inherited. To the extent that human action has created environmental features (e.g. global warming) it is common to talk of them being inherited, but this would normally be placed within niche construction or ecological inheritance, not necessarily cultural inheritance. More generally, I think the authors need to recognize that at one extreme their model does not concern culture at all – if their model found that transmission had no role, and all that mattered was environmental state, then the process at work is simply one of adaptive phenotypic plasticity and not culture. To resolve this the wording of the manuscript needs tweaking, particularly in this section, though the authors might spot other places to make useful changes too. The wording is great, for instance, in lines 100-101 where the point of the model is described as identifying the roles of transmission and ecology, and I would suggest that the latter is not part of culture per se (though it can of course influence culture).

line 73 – typo? “action facial unites”?

Reviewer #2: Thank you for addressing all the points raised in my first review. I have two minor comments on the revised manuscript and one optional additional suggestion.

Line 208: Why doesn’t the cultural trait converge to simply the default? In other words, why does rho still appear in the denominator?

Line 211: I hope it is a Wiener process and not a Wiener, an inhabitant of the city of Vienna. Or is it the *other* wiener and I completely misunderstood the point of transmitted culture? :D

Initial comment on the presentation of the model

The authors now explain all model parameters, but they have not addressed my central issue and neither improved the presentation of the model nor justified the design decisions. For example, why is cultural lability a necessary ingredient of the equation? Why do the external factors matter?

I don’t want to stay in the way of the paper getting published because I see the merit of the work. Still, I would encourage the authors to explain the generative story of their model first and then provide the equations, which will allow the readers to judge the model’s relevance for their problems and increase its adoption in the community. I leave it to the authors whether or not they want to address this point.

7. PLOS authors have the option to publish the peer review history of their article (what does this mean?). If published, this will include your full peer review and any attached files.

Reviewer #1: No

Reviewer #2: **Yes: **Peter Ranacher

---

## [Author Response · Author response to Decision Letter 1]

9 Feb 2022

Reviewer #1: The authors have revised and resubmitted their manuscript describing ETC models to evaluate the effects of cultural transmission and environmental factors on cultural change. I liked the paper on its initial submission and find it to be considerably improved following the authors’ revisions. I thank the authors for their work following the previous round of review, and I now consider the manuscript suitable for publication, although I will note two minor points:

line 32 – I appreciate the authors removal of the word institutions here, but there remains an issue with wording. As it stands, the authors now seem to be putting various features of the environment within “culture”, for instance pathogen presence. However, this is not typical. For instance, environmental features that arose independently of human actions might nonetheless influence how we behave, yet to include these features themselves as “culture” is unusual because it is not clear how they are being inherited. To the extent that human action has created environmental features (e.g. global warming) it is common to talk of them being inherited, but this would normally be placed within niche construction or ecological inheritance, not necessarily cultural inheritance. More generally, I think the authors need to recognize that at one extreme their model does not concern culture at all – if their model found that transmission had no role, and all that mattered was environmental state, then the process at work is simply one of adaptive phenotypic plasticity and not culture. To resolve this the wording of the manuscript needs tweaking, particularly in this section, though the authors might spot other places to make useful changes too. The wording is great, for instance, in lines 100-101 where the point of the model is described as identifying the roles of transmission and ecology, and I would suggest that the latter is not part of culture per se (though it can of course influence culture).

The reviewer is right when writing "To the extent that human action has created environmental features (e.g. global warming) it is common to talk of them being inherited, but this would normally be placed within niche construction or ecological inheritance, not necessarily cultural inheritance." Our model was not made to compare ecological inheritance and informational inheritance, but only ecology and transmitted information. The last sentence of the paragraph is thus not relevant here, and we have decided to remove it and replace it by:

“Here we are interested in quantifying the relative influence of evoked and transmitted forces onto cultural evolution.”

line 73 – typo? “action facial unites”?

Thank you for noting this, the correct expression is "facial action units" 

Ekman, P., & Rosenberg, E. L. (Eds.). (1997). What the face reveals: Basic and applied studies of spontaneous expression using the Facial Action Coding System (FACS). Oxford University Press, USA.

 

Reviewer #2: Thank you for addressing all the points raised in my first review. I have two minor comments on the revised manuscript and one optional additional suggestion.

Line 208: Why doesn’t the cultural trait converge to simply the default? In other words, why does rho still appear in the denominator?

This is simply a matter of the how the linear model is defined. Using the current definition, solving equation (6) without noise, ecological variable and connected region gives that the stable point is at -ρT_(t,r)+γ=0, i.e. T_(t,r)=ρ⁄γ.

We could have used another parametrization for equation (6) of the form (dT_(t,r))/dt=-ρ〖(T〗_(t,r)-γ ~)+⋯, in other words using the reparametrization γ ~=ρ⁄γ. Here this changed parameter γ ~ would be directly defined as the “default trait value”. The two systems are equivalent, really.

Line 211: I hope it is a Wiener process and not a Wiener, an inhabitant of the city of Vienna. Or is it the *other* wiener and I completely misunderstood the point of transmitted culture? :D

Thanks for pointing the missing word. That alternative hypothesis about the source of stochasticity in cultural evolution will be the subject of a subsequent paper.

Initial comment on the presentation of the model

The authors now explain all model parameters, but they have not addressed my central issue and neither improved the presentation of the model nor justified the design decisions. For example, why is cultural lability a necessary ingredient of the equation? Why do the external factors matter?

I don’t want to stay in the way of the paper getting published because I see the merit of the work. Still, I would encourage the authors to explain the generative story of their model first and then provide the equations, which will allow the readers to judge the model’s relevance for their problems and increase its adoption in the community. I leave it to the authors whether or not they want to address this point.

We thank the reviewer for helping us find how the clarity of the model presentation could be improved. We have made another attempt to try to improve it. It is about the best we can do, I am afraid. The following has been added to the manuscript (additions are underlined). 

We propose a simple stochastic linear model for the dynamics of cultural traits, where the degree of vertical transmission, horizontal transmission and evoked component (i.e. influence of the ecological variable) are controlled by separate parameters (to be estimated from the data). (…)

Parameter ρ is the cultural lability (or cultural leak). It determines the time scale at which the memory of cultural traits is lost, in other words it controls the degree of vertical transmission.

Because the raw value of cultural traits is usually arbitrary and one is usually mostly interested in relative values, one can often assume that γ =0, i.e. that the "default" trait value is null. If this is not the case, one must take caution in defining the generative process for artifacts to make sure that a bias term is not present that would make parameters not identifiable. For example, for binary artifacts, one could use p(A_(t,r) )=S(β_0+β_1 T_(t,r)) if γ is assumed to be null or p(A_(t,r) )=S(β_1 T_(t,r)) otherwise; the two systems are mathematically equivalent, they only differ in whether the constant bias β_0 is absorbed into a fixed offset of cultural traits or not.

The term “external factor” was not adapted, we actually referred to the ecological variable: λ controls the importance of the importance of the evoked component. This has been changed in the manuscript, and we hope it is clearer now.

---

## [Editor Report · Decision Letter 2]

14 Feb 2022

Evoked and Transmitted Culture models: Using bayesian methods to infer the evolution of cultural traits in history

PONE-D-21-13227R2

Dear Dr. HYAFIL,

We’re pleased to inform you that your manuscript has been judged scientifically suitable for publication and will be formally accepted for publication once it meets all outstanding technical requirements.

Kind regards,

Inés P. Mariño, Ph.D.

Academic Editor

PLOS ONE
---

## [Editor Report · Acceptance letter]

29 Mar 2022

PONE-D-21-13227R2 

Evoked and Transmitted Culture models: Using bayesian methods to infer the evolution of cultural traits in history 

Dear Dr. Hyafil:

I'm pleased to inform you that your manuscript has been deemed suitable for publication in PLOS ONE. Congratulations! Your manuscript is now with our production department. 

Kind regards, 

on behalf of

Dr. Inés P. Mariño 

Academic Editor

PLOS ONE